# Federated Learning with Reduced Information Leakage and Computation

**Tongxin Yin**[*]     *tyin@umich.edu*
*Department of Electrical and Computer Engineering*
*University of Michigan*

**Xuwei Tan**[*]     *tan.1206@osu.edu*
*Department of Computer Science and Engineering*
*The Ohio State University*

**Xueru Zhang**[*]     *zhang.12807@osu.edu*
*Department of Computer Science and Engineering*
*The Ohio State University*

**Mohammad Mahdi Khalili**     *khalili.17@osu.edu*
*Department of Computer Science and Engineering*
*The Ohio State University*

**Mingyan Liu**     *mingyan@umich.edu*
*Department of Electrical and Computer Engineering*
*University of Michigan*

**Reviewed on OpenReview:** *https://openreview.net/forum?id=ZJ4A3xhADV*

*These authors contributed equally to this work.

## Abstract

Federated learning (FL) is a distributed learning paradigm that allows multiple decentralized clients to collaboratively learn a common model without sharing local data. Although local data is not exposed directly, privacy concerns nonetheless exist as clients' sensitive information can be inferred from intermediate computations. Moreover, such information leakage accumulates substantially over time as the same data is repeatedly used during the iterative learning process. As a result, it can be particularly difficult to balance the privacy-accuracy trade-off when designing privacy-preserving FL algorithms. This paper introduces `Upcycled-FL`, a simple yet effective strategy that applies first-order approximation at every even round of model update. Under this strategy, half of the FL updates incur no information leakage and require much less computational and transmission costs. We first conduct the theoretical analysis on the convergence (rate) of `Upcycled-FL` and then apply two perturbation mechanisms to preserve privacy. Extensive experiments on both synthetic and real-world data show that the `Upcycled-FL` strategy can be adapted to many existing FL frameworks and consistently improve the privacy-accuracy trade-off.[1]

## 1 Introduction

Federated learning (FL) has emerged as an important paradigm for learning models in a distributed fashion, whereby data is distributed across different clients and the goal is to jointly learn a model from the distributed data. This is facilitated by a central server and the model can be learned through iterative interactions

---

[1]Code available at https://github.com/osu-srml/Upcycled-FL.

between the central server and clients: at each iteration, each client performs certain computation using its local data; the updated local models are collected and aggregated by the server; the aggregated model is then sent back to clients for them to update local models; and so on till the learning task is deemed accomplished.

Although each client's data is not shared directly with the central server, there is a risk of information leakage that a third party may infer individual sensitive information from (intermediate) computational outcomes. Information leakage occurs whenever the client's local gradients are shared with third parties. Importantly, the information leakage (or privacy loss) **accumulates** as data is repeatedly used during the iterative learning process: with more computational outcomes derived from individual data, third parties have more information to infer sensitive data and it poses higher privacy risks for individuals. An example is Huang et al. (2021), which shows that eavesdroppers can conduct gradient inversion attacks to recover clients' data from the gradients. In this paper, we use differential privacy (DP) proposed by Dwork (2006), a de facto standard for preserving individual data privacy in data analysis, ranging from simple tasks such as data collection and statistical analysis (Zhang et al., 2022b; Ghosh & Roth, 2011; Khalili & Vakilinia, 2021; Amin et al., 2019; Liu et al., 2021; Khalili et al., 2021c; 2019) to complex machine learning and optimization tasks (Cai et al., 2024; Khalili et al., 2021b;a; Huang et al., 2020; Jagielski et al., 2019; Zhang et al., 2019b; 2018a;b; 2019c). Compared to other privacy preservation techniques, it can (i) rigorously quantify the total privacy leakage for complex algorithms such as FL; (ii) defend against attackers regardless of their background knowledge; (iii) provide heterogeneous privacy guarantees for different clients. To achieve a certain DP guarantee, we need to perturb FL algorithms (e.g., adding noise to the output, objective, or gradient of local clients) and the perturbation needed for achieving the privacy requirement of each client grows as the total information leakage increases. Because the added perturbation inevitably reduces algorithm accuracy, it can be difficult to balance the privacy and accuracy trade-off in FL.

This paper proposes a novel strategy for FL called Upcycled Federated Learning (`Upcycled-FL`)[2], in which clients' information leakage can be reduced such that it only occurs during *half* of the updates. This is attained by modifying the *even* iterations of a baseline FL algorithm with first-order approximation, which allows us to update the model using existing model parameters from previous iterations without using the client's data. Moreover, the updates in even iterations only involve addition/subtraction operations on existing model parameters at the central server. Because Upcycled-FL doesn't require local training and transmission in half of iterations, the transmission costs and the training time can be reduced significantly. It turns out that `Upcycled-FL`, by reducing the total information leakage, requires less perturbation to attain a certain level of privacy and can enhance the privacy-accuracy trade-off significantly.

We emphasize that the idea of "upcycling information" is orthogonal to both the baseline FL algorithm and the DP perturbation method. It can be applied to any FL algorithms that involve local optimization at the clients. In this paper, we apply `Upcycled-FL` strategy to multiple existing FL algorithms and evaluate them on both synthetic and real-world datasets. For DP perturbation methods, we consider both output and objective perturbation as examples to quantify the privacy loss, while other DP methods can also be used.

It is worth noting that although differentially private federated learning has been extensively studied in the literature, e.g., (Asoodeh et al., 2021; Chuanxin et al., 2020; Zhang et al., 2022a; Zheng et al., 2021; Wang et al., 2020b; Kim et al., 2021; Zhang et al., 2019a; Baek et al., 2021; Wu et al., 2022; Girgis et al., 2021; Truex et al., 2020; Hu et al., 2020; Seif et al., 2020; Zhao et al., 2020; Wei et al., 2020; Triastcyn & Faltings, 2019), all these algorithms need client's local data to update the model and the information leakage inevitably occurs at every iteration. This is fundamentally different from this work, where we propose a novel strategy that effectively reduces information leakage in FL.

In addition to private federated learning, several approaches were proposed in the DP literature to improve the privacy-accuracy trade-off, e.g., privacy amplification by *sampling* (Balle et al., 2018; Beimel et al., 2014; Hu et al., 2021; Wang et al., 2019; Kasiviswanathan et al., 2011; Wang et al., 2015; Abadi et al., 2016), *leveraging non-private public data* (Avent et al., 2017; Papernot et al., 2016), *shuffling* (Úlfar Erlingsson et al., 2019), *using weaker privacy notion* (Bun & Steinke, 2016), *using tighter privacy composition analysis tool* (Abadi et al., 2016). However, none of these strategies affect the algorithmic properties of learning algorithms. By contrast, our method improves the privacy-accuracy trade-off by modifying the property of FL algorithm

---

[2]The word "upcycle" refers to reusing material so as to create higher-quality things than the original.

(i.e., reducing the total information leakage); this improvement on the algorithmic property is independent of the privacy notion/mechanism or the analysis method.

Our main contributions are summarized as follows.

- We propose `Upcycled-FL` (Algorithm 1), a novel strategy with reduced information leakage and computation that is broadly applicable to many existing FL algorithms.

- As an example, we apply our strategy to `FedProx` (Li et al., 2020) and conduct convergence (rate) analysis (Section 5, Theorem 5.6) where we identify a sufficient condition for the convergence of `Upcycled-FL`.

- As an example, we apply two differential privacy mechanisms (i.e., output perturbation and objective perturbation) to `Upcycled-FL` and conduct privacy analysis (Section 6, Theorem 6.2).

- We evaluate the effectiveness of `Upcycled-FL` on both synthetic and real data (Section 7). Extensive experiments show that `Upcycled-FL` can be adapted to many existing federated algorithms to achieve better performance; it effectively improves the accuracy-privacy trade-off by reducing information leakage.

## 2   Related Work

**Differential privacy in federated learning.** Differential privacy has been widely used in federated learning to provide privacy guarantees (Asoodeh et al., 2021; Chuanxin et al., 2020; Zhang et al., 2022a; Zheng et al., 2021; Wang et al., 2020b; Kim et al., 2021; Zhang et al., 2019a; Baek et al., 2021; Wu et al., 2022; Girgis et al., 2021; Truex et al., 2020; Hu et al., 2020; Seif et al., 2020; Zhao et al., 2020; Wei et al., 2020; Triastcyn & Faltings, 2019). For example, Zhang et al. (2022a) uses the Gaussian mechanism for a federated learning problem and propose an incentive mechanism to encourage users to share their data and participate in the training process. Zheng et al. (2021) introduces $f$-differential privacy, a generalized version of Gaussian differential privacy, and propose a federated learning algorithm satisfying this new notion. Wang et al. (2020b) proposes a new mechanism called Random Response with Priori (RRP) to achieve local differential privacy and apply this mechanism to the text data by training a Latent Dirichlet Allocation (LDA) model using a federated learning algorithm. Triastcyn & Faltings (2019) adapts the Bayesian privacy accounting method to the federated setting and propose joint accounting method for estimating client-level and instance-level privacy simultaneously and securely. Wei et al. (2020) presents a private scheme that adds noise to parameters at the random selected devices before aggregating and provides a convergence bound. Kim et al. (2021) combines the Gaussian mechanism with gradient clipping in federated learning to improve the privacy-accuracy tradeoff. Asoodeh et al. (2021) considers a different setting where only the last update is publicly released and the central server and other devices are assumed to be trustworthy. However, all these algorithms need client's local data to update the model and the information leakage inevitably occurs at every iteration. This is fundamentally different from `Upcycled-FL`, which reuses existing results to update half of iterations and significantly reduces information leakage and computation.

**Tackling heterogeneity in federated learning.** It's worth mentioning that, `Upcycled-FL` also empirically outperforms existing baseline algorithms under device and statistical heterogeneity. In real-world scenarios, local data are often non-identically distributed across different devices; different devices are also often equipped with different specifications and computation capabilities. Such heterogeneity often causes instability in the model performance and leads to divergence. Many approaches have been proposed to tackle this issue in FL. For example, `FedAvg` (McMahan et al., 2017) uses a random selection of devices at each iteration to reduce the negative impact of statistical heterogeneity; however, it may fail to converge when heterogeneity increases. Other methods include `FedProx` (Li et al., 2020), a generalization and re-parameterization of `FedAvg` that adds a proximal term to the objective function to penalize deviations in the local model from the previous aggregation, and `FedNova` (Wang et al., 2020a) that re-normalizes local updates before updating to eliminate objective inconsistency. It turns out that `Upcycled-FL` exhibits superior performance in the presence of heterogeneity because gradients encapsulate information on data heterogeneity, the reusing of which leads to a boost in performance.

## 3 Problem Formulation

Consider an FL system consisting of a central server and a set $\mathcal{I}$ of clients. Each client $i$ has its local dataset $\mathcal{D}_i$ and these datasets can be non-i.i.d across the clients. The goal of FL is to learn a model $\omega \in \mathbb{R}^d$ from data $\cup_{i \in \mathcal{I}} \mathcal{D}_i$ by solving the following optimization:

$$\min_\omega f(\omega) := \sum_{i \in \mathcal{I}} p_i F_i(\omega; \mathcal{D}_i) = \mathbb{E}\left[F_i(\omega; \mathcal{D}_i)\right], \tag{1}$$

where $p_i = \frac{|\mathcal{D}_i|}{\sum_{j \in \mathcal{I}} |\mathcal{D}_j|}$ is the size of client $i$'s data as a fraction of the total data samples, $\mathbb{E}[\cdot]$ is defined as the expectation over clients, $F_i(\omega; \mathcal{D}_i)$ is the local loss function associated with client $i$ and depends on local dataset $\mathcal{D}_i$. In this work, we allow $F_i(\omega; \mathcal{D}_i)$ to be possibly non-convex.

**FL Algorithm.** Let $\omega_i^t$ be client $i$'s local model parameter at time $t$. In FL, the model is learned through an iterative process: at each time step $t$, 1) *local computations:* each active client updates its local model $\omega_i^t$ using its local data $\mathcal{D}_i$; 2) *local models broadcasts:* local models (or gradients) are then uploaded to the central server; 3) *model aggregation:* the central server aggregates results received from clients to update the global model parameter $\overline{\omega}^t = \sum_{i \in \mathcal{I}} p_i \omega_i^t$; 4) *model updating:* the aggregated model is sent back to clients and is used for updating local models at $t + 1$.

During the learning process, each client's local computation is exposed to third parties at every iteration: its models/gradients need to be uploaded to the central server, and the global models calculated based on them are shared with all clients. It is thus critical to ensure the FL is privacy-preserving. In this work, we consider differential privacy as the notion of privacy.

**Differential Privacy (Dwork, 2006).** Differential privacy (DP) centers around the idea that the output of a certain computational procedure should be statistically similar given singular changes to the input, thereby preventing meaningful inference from observing the output.

In FL, the information exposed by each client $i$ includes all intermediate computations $\{\omega_i^t\}_{t=1}^T$. Consider a randomized FL algorithm $\mathcal{A}(\cdot)$ that generates a sequence of private local models $\{\widehat{\omega}_i^t\}_{t=1}^T$, we say it satisfies $(\varepsilon, \delta)$-differential privacy for client $i$ over $T$ iterations if the following holds for any possible output $O \in \mathbb{R}^d \times \cdots \times \mathbb{R}^d$, and for any two neighboring local datasets $\mathcal{D}_i$, $\mathcal{D}_i'$:

$$\Pr(\{\widehat{\omega}_i^t\}_{t=0}^T \in O | \mathcal{D}_i) \leq \exp(\varepsilon) \cdot \Pr(\{\widehat{\omega}_i^t\}_{t=0}^T \in O | \mathcal{D}_i') + \delta.$$

where $\varepsilon \in [0, \infty)$ bounds the privacy loss, and $\delta \in [0, 1]$ loosely corresponds to the probability that algorithm fails to bound the privacy loss by $\varepsilon$. Two datasets are neighboring datasets if they are different in at most one data point.

## 4 Proposed Method: `Upcycled-FL`

**Main idea.** Fundamentally, the accumulation of information leakage over iterations stems from the fact that the client's data $\mathcal{D}_i$ is used in every update. If the updates can be made without directly using this original data, but only from computational results that already exist, then the information leakage originating from these updates will be zero, and meanwhile, the computational cost may be reduced significantly. Based on this idea, we propose `Upcycled-FL`, which considers reusing the earlier computations in a new update and significantly reduces total information leakage and computational cost. Note that `Upcycled-FL` is not a specific algorithm but an effective strategy that can be used for any existing FL algorithms.

**Upcycling model update.** Next, we present `Upcycled-FL` and illustrate how the client's total information leakage is reduced under this method.

For an FL system with the objective shown in Eqn. (1), the client $i$'s local objective function is given by $F_i(\omega; \mathcal{D}_i)$. Under `Upcycled-FL`, we apply *first-order approximation* to $F_i(\omega; \mathcal{D}_i)$ at **even** iterations during federated training (while odd updates remain fixed). Specifically, at $2m$-th iteration, we expand $F_i(\omega; \mathcal{D}_i)$ at

$\omega_i^{2m-1}$ (local model in the previous iteration). Based on the Taylor series expansion, we have:

$$
\begin{aligned}
F_i(\omega; \mathcal{D}_i) &= F_i(\omega_i^{2m-1}; \mathcal{D}_i) + \nabla F_i(\omega_i^{2m-1}; \mathcal{D}_i)^T(\omega - \omega_i^{2m-1}) + \mathcal{O}(||\omega - \omega_i^{2m-1}||^2) \\
&\approx F_i(\omega_i^{2m-1}; \mathcal{D}_i) + \nabla F_i(\omega_i^{2m-1}; \mathcal{D}_i)^T(\omega - \omega_i^{2m-1}) + \frac{\lambda_m}{2}||\omega - \omega_i^{2m-1}||^2
\end{aligned}
\tag{2}
$$

for some constant $\lambda_m \geq 0$ which may differ for different iteration $2m$. Then for an FL algorithm, its model update at $2m$-th iteration under `Upcycled-FL` strategy can be attained by replacing $F_i(\omega; \mathcal{D}_i)$ with its approximation in Eqn. (2) while the updates at odd iterations remain the same. We illustrate this using the following two examples.

**Example 4.1** (`FedAvg` (McMahan et al., 2017) under `Upcycled-FL` strategy). *In FedAvg, client $i$ at each iteration updates the local model by optimizing its local objective function, i.e., $\omega_i^t = \arg\min_\omega F_i(\omega; \mathcal{D}_i), \forall t$. Under Upcycled-FL strategy, the client $i$'s updates become:*

$$
\omega_i^t = \begin{cases} \arg\min_\omega \nabla F_i(\omega_i^{2m-1}; \mathcal{D}_i)^T \omega + \frac{\lambda_m}{2}||\omega - \omega_i^{2m-1}||^2, & if \quad t = 2m \\ \arg\min_\omega F_i(\omega; \mathcal{D}_i), & if \quad t = 2m-1 \end{cases}
$$

**Example 4.2** (`FedProx` (Li et al., 2020) under `Upcycled-FL` strategy). *In FedProx, a proximal term is added to the local objective function to stabilize the algorithm under heterogeneous clients (Algorithm 2), i.e., client $i$ at each iteration updates the local model $\omega_i^t = \arg\min_\omega F_i(\omega; \mathcal{D}_i) + \frac{\mu}{2}||\omega - \overline{\omega}^{t-1}||^2, \forall t$. Under Upcycled-FL strategy, the client $i$'s updates become:*

$$
\omega_i^t = \begin{cases} \arg\min_\omega \nabla F_i(\omega_i^{2m-1}; \mathcal{D}_i)^T \omega + \frac{\lambda_m}{2}||\omega - \omega_i^{2m-1}||^2 + \frac{\mu}{2}||\omega - \overline{\omega}^{2m-1}||^2, & if \quad t = 2m \\ \arg\min_\omega F_i(\omega; \mathcal{D}_i) + \frac{\mu}{2}||\omega - \overline{\omega}^{2m-2}||^2, & if \quad t = 2m-1 \end{cases}
\tag{3}
$$

Next, we demonstrate how the information is upcycled under the above idea. As an illustrating example, we focus on `FedProx` given in Example 4.2.

Note that in the even update of Eqn. (3), the only term that depends on dataset $\mathcal{D}_i$ is $\nabla F_i(\omega_i^{2m-1}; \mathcal{D}_i)$, which can be derived directly from the previous odd iteration. Specifically, according to the first-order condition, the following holds at odd iterations:

$$
\omega_i^{2m-1} = \arg\min_\omega F_i(\omega; \mathcal{D}_i) + \frac{\mu}{2}||\omega - \overline{\omega}^{2m-2}||^2 \xRightarrow{\text{imply}} \nabla F_i(\omega_i^{2m-1}; \mathcal{D}_i) + \mu(\omega_i^{2m-1} - \overline{\omega}^{2m-2}) = 0.
\tag{4}
$$

Plug $\nabla F_i(\omega_i^{2m-1}; \mathcal{D}_i)$ into even update of (3), we have the estimated update from the odd update:

$$
\omega_i^{2m} \approx \arg\min_\omega \mu(\overline{\omega}^{2m-2} - \omega_i^{2m-1})^T \omega + \frac{\lambda_m}{2}||\omega - \omega_i^{2m-1}||^2 + \frac{\mu}{2}||\omega - \overline{\omega}^{2m-1}||^2.
\tag{5}
$$

where the approximately equals sign "$\approx$" is due to the approximation in Eqn. (2). By first-order condition, even update (5) can be reduced to:

$$
\omega_i^{2m} \approx \omega_i^{2m-1} + \frac{\mu}{\mu + \lambda_m}\left(\overline{\omega}^{2m-1} - \overline{\omega}^{2m-2}\right).
\tag{6}
$$

It turns out that with first-order approximation, dataset $\mathcal{D}_i$ is not used in the even updates. Because these updates do not require access to the client's data, these updates can be conducted at the central server directly. That is, the central server updates the global model by aggregating:

$$
\overline{\omega}^{2m} = \sum_{i \in \mathcal{I}} p_i \omega_i^{2m} \approx \sum_{i \in \mathcal{I}} p_i \omega_i^{2m-1} + \frac{\mu}{\mu + \lambda_m}\left(\overline{\omega}^{2m-1} - \overline{\omega}^{2m-2}\right) = \overline{\omega}^{2m-1} + \frac{\mu}{\mu + \lambda_m}\left(\overline{\omega}^{2m-1} - \overline{\omega}^{2m-2}\right)
$$

Therefore, under `Upcycled-FL` strategy, even updates only involve *addition/subtraction* operations on the existing global models from previous iterations (i.e., $\overline{\omega}^{2m-1}, \overline{\omega}^{2m-2}$) without the need for a local training epoch: both the computational cost and transmission cost are reduced significantly. Note that the first-order

approximation is only applied to even iterations, while the odd iterations should remain as the origin to ensure Eqn. (4) holds. The entire updating procedure of `Upcycled-FL` is summarized in Algorithm 1.

Because $\mathcal{D}_i$ is only used in odd iterations, information leakage only happens during odd updates. Intuitively, the reduced information leakage would require less perturbation to attain a certain level of privacy guarantee, which further results in higher accuracy and improved privacy-accuracy trade-off. In the following sections, we first analyze the convergence property of `Upcycled-FL` and then apply privacy mechanisms to satisfy differential privacy.

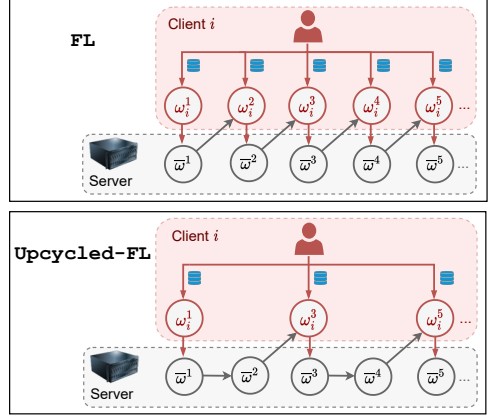
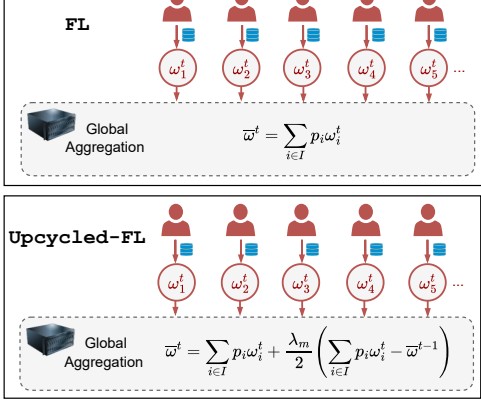

(a) `Upcycled-FL` reuses the intermediate updates     (b) `Upcycled-FL` uses different aggregation rule

Figure 1: `Upcycled-FL` can be considered from two perspectives: (i) it can be regarded as reusing the intermediate updates of local models to reduce the total information leakage; or (ii) it can be regarded as a global aggregation method with larger global update, which accelerates the learning process with the same information leakage under the same training iterations.

**Discussion.** Indeed, if we view two consecutive steps (odd $2m-1$ followed by even $2m$) of `Upcycled-FL` as a single iteration $t$, then `Upcycled-FedProx` and `FedProx` will incur the same information leakage but will differ at the phase of *global model aggregation*, as shown in Figure 1(b). Specifically, instead of simply aggregating the global model by averaging over local updates (i.e., $\sum_{i\in\mathcal{I}} p_i \omega_i^t$), `Upcycled-FL` not only takes the average of local updates but also pushes the aggregation moving more toward the updating direction (i.e., $\sum_{i\in\mathcal{I}} p_i \omega_i^t - \overline{\omega}^{t-1}$). We present the difference between `Upcycled-FL` update strategy and regular aggregation strategy from these two perspectives in Figure 1. As `Upcycled-FL` only accesses client data at even iterations, it halves the communication cost compared to standard FL methods with the same number of iterations.

---

**Algorithm 1** Proposed aggregation strategy: `Upcycled-FL`

---

1: **Input:** $\lambda_m > 0$, $\mu > 0$, $\{\mathcal{D}_i\}_{i\in\mathcal{I}}$, $\overline{\omega}^0$
2: **for** $m = 1$ **to** $M$ **do**
3:     The central server sends the global model parameters $\overline{\omega}^{2m-2}$ to all the clients.
4:     A subset of clients are selected to be active and each active client updates its local model by finding an (approximate) minimizer of the local objective function:

$$\omega_i^{2m-1} \leftarrow \arg\min_{\omega} F_i(\omega; \mathcal{D}_i) \qquad \text{or} \quad \textit{other local objective functions}$$

5:     Clients send local models to the central server.
6:     The central server updates the global model by aggregating all local models:

$$
\begin{aligned}
\overline{\omega}^{2m-1} &= \sum_{i\in\mathcal{I}} p_i \omega_i^{2m-1} \\
\overline{\omega}^{2m} &= \overline{\omega}^{2m-1} + \frac{\lambda_m}{2}\left(\overline{\omega}^{2m-1} - \overline{\omega}^{2m-2}\right)
\end{aligned}
$$

7: **end for**

---

# 5 Convergence Analysis

In this section, we analyze the convergence of `Upcycled-FL`. For illustrating purposes, we focus on analyzing the convergence of `Upcycled-FedProx`. Note that we do not require local functions $F_i(\cdot)$ to be convex. Moreover, we consider practical settings where data are *non-i.i.d* across different clients. Similar to Li et al. (2020), we introduce a measure below to quantify the dissimilarity between clients in the federated network.

**Definition 5.1** (*B*-Dissimilarity (Li et al., 2020))**.** The local loss function $F_i$ is *B*-dissimilar if $\forall \omega$, we have $\mathbb{E}[||\nabla F_i(\omega)||^2] \leq ||\nabla f(\omega)||^2 B^2$.

where $\mathbb{E}[\cdot]$ denotes the expectation over clients (see Eqn. (1)). Parameter $B \geq 1$ captures the statistical heterogeneity across different clients: when all clients are homogeneous with i.i.d. data, we have $B = 1$ for all local functions; the larger value of $B$, the more dissimilarity among clients.

**Assumption 5.2.** Local loss functions $F_i$ are *B*-dissimilar and *L*-Lipschitz smooth.

Note that *B*-dissimilarity can be satisfied if the divergence between the gradient of the local loss function and that of the aggregated global function is bounded, as stated below.

**Lemma 5.3.** $\forall i$, there exists $B$ such that $F_i$ is *B*-dissimilar if $||\nabla F_i(\omega) - \nabla f(\omega)|| \leq \kappa_i, \forall \omega$ for some $\kappa_i$.

**Assumption 5.4.** $\forall i$, $h_i(\omega; \overline{\omega}^t) := F_i(\omega; \mathcal{D}_i) + \frac{\mu}{2}||\omega - \overline{\omega}^t||^2$ are $\rho$-strongly convex.

The above assumptions are fairly standard. They first appeared in Li et al. (2020) and are adopted in subsequent works such as T Dinh et al. (2020); Khaled et al. (2020); Pathak & Wainwright (2020). Note that strongly convex assumption is not on local objective $F_i(\omega; \mathcal{D}_i)$, but the regularized function $F_i(\omega; \mathcal{D}_i) + \frac{\mu}{2}||\omega - \overline{\omega}^t||^2$, i.e., the assumption can be satisfied by selecting a sufficiently large $\mu > 0$. Indeed, as shown in Section 7, our algorithm still converges even when Assumption 5.2 and 5.4 do not hold (e.g., DNN). Next, we conduct the theoretical analysis on the convergence rate.

**Lemma 5.5.** *Let $\mathcal{S}_m$ be a set of $K$ randomly selected clients which got updated (i.e., active clients) at iterations $2m-1$ and $2m$, and $\mathbb{E}_{\mathcal{S}_m}[\cdot]$ be the expectation with respect to the choice of clients. Then under Assumption 5.2 and 5.4, we have*

$$\mathbb{E}_{\mathcal{S}_m}[f(\overline{\omega}^{2m+1})] \leq f(\overline{\omega}^{2m-1}) - \boldsymbol{C_1}||\nabla f(\overline{\omega}^{2m-1})||^2 + \boldsymbol{C_2} \cdot h_m^1 + \boldsymbol{C_3} \cdot h_m^2,$$

*where*

$$\begin{aligned} h_m^1 &:= ||\nabla f(\overline{\omega}^{2m-1})|| \cdot ||\overline{\omega}^{2m-1} - \overline{\omega}^{2m-2}||; \\ h_m^2 &:= ||\overline{\omega}^{2m-1} - \overline{\omega}^{2m-2}||^2. \end{aligned}$$

*The details of term $\boldsymbol{C_1}$, $\boldsymbol{C_2}$, $\boldsymbol{C_3}$ (expressed as functions of $L, B, \frac{1}{\mu}, \frac{1}{\rho}, \frac{1}{K}, \frac{\mu}{\lambda_m}$) are in Appendix C, Eqn. (11)-(13).*

Lemma 5.5 characterizes the relation of the values of global objective function over *two consecutive odd iterations*. It is easy to verify $\boldsymbol{C_2}, \boldsymbol{C_3} \geq 0$. By rearranging and telescoping, we get the following convergence rate of `Upcycled-FedProx`.

**Theorem 5.6** (Convergence rate of `Upcycled-FedProx`)**.** *Under Assumption 5.2 and 5.4, if $\boldsymbol{C_1} > 0$, we have*

$$\begin{aligned} \min_{m \in [M]} \mathbb{E}\left[||\nabla f(\overline{\omega}^{2m-1})||^2\right] &\leq \frac{1}{M}\sum_{m=0}^{M} \mathbb{E}\left[||\nabla f(\overline{\omega}^{2m-1})||^2\right] \\ &\leq \frac{f(\overline{\omega}^0) - f(\overline{\omega}^*)}{M\boldsymbol{C_1}} + \frac{\sum_{m=0}^{M}\boldsymbol{C_2}h_m^1}{M\boldsymbol{C_1}} + \frac{\sum_{m=0}^{M}\boldsymbol{C_3}h_m^2}{M\boldsymbol{C_1}}, \end{aligned}$$

*where $\overline{\omega}^0$ and $\overline{\omega}^*$ denote the initial and the optimal global model parameters, respectively. Both terms $\boldsymbol{C_2}$ and $\boldsymbol{C_3}$ are decreasing in $\frac{\lambda_m}{\mu}$.*

Theorem 5.6 implies that tunable $\mu$ and $\lambda_m$ are key hyper-parameters that control the convergence (rate) and robustness of `Upcycled-FedProx`. Recall that $\mu$ penalizes the deviation of local model $\omega_i^{2m}$ from global aggregated model $\overline{\omega}^{2m-1}$, while $\lambda_m$ penalizes the update of local model $\omega_i^{2m}$ from its previous update $\omega_i^{2m-1}$.

Because $\boldsymbol{C_1} := C_1\left(L, B, \frac{1}{\mu}, \frac{1}{\rho}, \frac{1}{K}\right)$ does not depend on $\lambda_m$ (by Eqn. (11)), for proper $\mu$ and local functions $F_i$, the condition $\boldsymbol{C_1} > 0$ in Theorem 5.6 can hold for any $\lambda_m$. However, $\lambda_m$ could affect the convergence rate via terms $\boldsymbol{C_2} := C_2\left(L, B, \frac{1}{\mu}, \frac{1}{\rho}, \frac{1}{K}, \frac{\mu}{\mu+\lambda_m}\right)$ and $\boldsymbol{C_3} := C_3\left(L, \frac{1}{\mu}, \frac{1}{\rho}, \frac{1}{K}, \frac{\mu}{\mu+\lambda_m}\right)$. Specifically, as the ratio $\frac{\lambda_m}{\mu}$ increases, both $\boldsymbol{C_2}$ and $\boldsymbol{C_3}$ decrease (by Eqn. (12)-(13)) which results in a tighter convergence rate bound. We empirically examine the impacts of $\mu$ and $\lambda_m$ in Section 7.

It is worth noting that the convergence rate also depends on data heterogeneity, captured by dissimilarity $B$. According to Eqn. (11), $\boldsymbol{C_1} > 0$ must hold when $B = 0$ (i.i.d. clients). Although $\boldsymbol{C_1}$ may become negative as $B$ increases, the experiments in Section 7 show that `Upcycled-FedProx` can still converge when data is highly heterogeneous.

**Assumption 5.7.** $||\overline{\omega}^{2m-1} - \overline{\omega}^{2m-2}|| \le h, \forall m$ and $||\nabla f(\omega)|| \le d, \forall \omega$.

Assumption 5.7 is standard and has been used when proving the convergence of FL algorithms (Li et al., 2019; Yang et al., 2022); it requires that the difference of aggregated weights between two consecutive iterations and the gradient $||\nabla f(\omega)||$ are bounded. Under this assumption, we have the following corollary.

**Corollary 5.8** (Convergence to the stationary point). *Under Assumption 5.2, 5.4, and 5.7, for fixed $\mu, K$, if $\lambda_m$ is taken such that $\frac{\mu}{\mu+\lambda_m} = \mathcal{O}\left(\frac{1}{\sqrt{M}}\right)$, then the convergence rate of* `Upcycled-FedProx` *reduces to $\mathcal{O}(\frac{1}{\sqrt{M}})$.*

Corollary 5.8 provides guidance on selecting the value of $\lambda_m$ properly to guarantee the convergence of `Upcycled-FedProx`, i.e., by taking an increasing sequence of $\{\lambda_m\}_{m=1}^M$. Intuitively, increasing $\lambda_m$ during the training helps stabilize the algorithm, because the deviation of local models from the previous update gets penalized more under a larger $\lambda_m$.

# 6 Private `Upcycled-FL`

In this section, we present a privacy-preserving version of `Upcycled-FL`. Many perturbation mechanisms can be adopted to achieve differential privacy such as *objective perturbation* (Chaudhuri et al., 2011; Kifer et al., 2012), *output perturbation* (Chaudhuri et al., 2011; Zhang et al., 2017), *gradient perturbation* (Bassily et al., 2014; Wang et al., 2017), etc. In this section, we use output and objective perturbation as examples to illustrate that FL algorithms combined with `Upcycled-FL` strategy, by reducing the total information leakage, can attain a better privacy-accuracy trade-off. Note that both output and objective perturbation methods are used to generate private updates at odd iterations, which can be used directly for even updates.

*Output perturbation:* the private odd updates $\widehat{\omega}_i^{2m-1}$ are generated by first *clipping* the local models $\omega_i^{2m-1}$ and then adding a noise random vector $n_i^m$ to the clipped model:

$$\textbf{\textit{Clip odd update:}} \quad \xi(\omega_i^{2m-1}) = \frac{\omega_i^{2m-1}}{\max\left(1, \frac{||\omega_i^{2m-1}||_2}{\tau}\right)}$$

$$\textbf{\textit{Perturb with noise:}} \quad \widehat{\omega}_i^{2m-1} = \xi(\omega_i^{2m-1}) + n_i^m$$

where parameter $\tau > 0$ is the clipping threshold; the clipping ensures that if $||\omega_i^{2m-1}||_2 \le \tau$, then update remains the same, otherwise it is scaled to be of norm $\tau$.

*Objective perturbation:* a random linear term $\langle n_i^m, \omega \rangle$ is added to the objective function in odd $(2m+1)$-th iteration, and the private local model $\widehat{\omega}_i^{2m+1}$ is found by solving a *perturbed* optimization.

Taking `Upcycled-FedProx` as the example, we have:

$$\widehat{\omega}_i^{2m+1} = \arg\min_\omega F_i(\omega; \mathcal{D}_i) + \frac{\mu}{2}||\omega - \overline{\omega}^{2m}||^2 + \langle n_i^m, \omega \rangle,$$

Given noisy $\widehat{\omega}_i^{2m-1}$ generated by either method, the private even updates $\widehat{\omega}_i^{2m}$ can be computed directly at the central server using the noisy aggregation $\sum_{i \in \mathcal{I}} p_i \widehat{\omega}_i^{2m-1}$.

**Privacy Analysis.** Next, we conduct privacy analysis and theoretically quantify the total privacy loss of private `Upcycled-FL`. Because even updates are computed directly using already private intermediate results

without using dataset $\mathcal{D}_i$, no privacy leakage occurs at even iterations. This can be formally stated as the following lemma.

**Lemma 6.1.** *For any $m = 1, 2, \cdots$, if the total privacy loss up to $2m - 1$ can be bounded by $\varepsilon_m$, then the total privacy loss up to the $2m$-th iteration can also be bounded by $\varepsilon_m$.*

Lemma 6.1 is straightforward; it can be derived directly by leveraging a property of differential privacy called *immunity to post-processing* (Dwork et al., 2014), i.e., a differentially private output followed by any data-independent computation remains satisfying differential privacy.

Based on Lemma 6.1, we can quantify the total privacy loss of private `Upcycled-FL`. We shall adopt moments accountant method (Abadi et al., 2016) for output perturbation, and the analysis method in (Chaudhuri et al., 2011; Zhang & Zhu, 2016; Zhang et al., 2018b) for objective perturbation.

Here, we focus on settings where local loss function $F_i(\omega_i, \mathcal{D}_i) := \frac{1}{|\mathcal{D}_i|} \sum_{d \in \mathcal{D}_i} \hat{F}_i(\omega_i, d)$ for some $\hat{F}_i$, and the guarantee of privacy is presented in Theorem 6.2 (output perturbation) and 6.3 (objective perturbation) below. The total privacy loss in the following theorem is composed using moments accountant method (Abadi et al., 2016).

**Theorem 6.2.** *Consider the private `Upcycled-FL` over $2M$ iterations under output perturbation with noise $n_i^m \sim \mathcal{N}(0, \sigma^2 \mathbf{I})$, then for any $\varepsilon \geq \frac{M\tau^2}{2\sigma^2 |\mathcal{D}_i|^2}$, the algorithm is $(\varepsilon, \delta)$-DP for agent $i$ for*

$$\delta = \exp\left(-\frac{M\tau^2}{2\sigma^2 |\mathcal{D}_i|^2} \left(\frac{\varepsilon \sigma^2 |\mathcal{D}_i|^2}{M\tau^2} - \frac{1}{2}\right)^2\right).$$

*Equivalently, for any $\delta \in [0, 1]$, the algorithm is $(\varepsilon, \delta)$-DP for agent $i$ for*

$$\varepsilon = 2\sqrt{\frac{M\tau^2}{2\sigma^2 |\mathcal{D}_i|^2} \log(\frac{1}{\delta})} + \frac{M\tau^2}{2\sigma^2 |\mathcal{D}_i|^2}.$$

**Theorem 6.3.** *Consider the private `Upcycled-FL` over $2M$ iterations under objective perturbation with noise $n_i^m \sim \exp\left(-\alpha_i^m ||n_i^m||_2\right)$. Suppose $\hat{F}_i$ is generalized linear model (Iyengar et al., 2019; Bassily et al., 2014)[3] that satisfies $||\nabla \hat{F}_i(\omega; d)|| < u_1$, $|\hat{F}_i''| \leq u_2$. Let feature vectors be normalized such that its norm is no greater than 1, and suppose $u_2 \leq 0.5 |\mathcal{D}_i| \mu$ holds. Then the algorithm satisfies $(\varepsilon, 0)$-DP for agent $i$ where $\varepsilon = \sum_{m=0}^{M} \frac{2\alpha_i^m u_1 \mu + 2.8 u_2}{|\mathcal{D}_i| \mu}$.*

The assumptions on $\hat{F}_i$ are again fairly standard in the literature, see e.g., (Chaudhuri et al., 2011; Zhang & Zhu, 2016; Zhang et al., 2018b). Theorem 6.2 and 6.3 show that the total privacy loss experienced by each agent accumulates over iterations and privacy loss only comes from odd iterations. In contrast, if consider differentially private `FedProx`, accumulated privacy loss would come from all iterations. Therefore, to achieve the same privacy guarantee, private `Upcycled-FL` requires much less perturbation per iteration than private `FedProx`. As a result, accuracy can be improved significantly. Experiments in Section 7 show that `Upcycled-FL` significantly improves privacy-accuracy trade-off compared to other methods.

# 7 Experiments

In this section, we empirically evaluate the performance of `Upcycled-FL` by combining it with several popular FL methods. We first consider non-private algorithms to examine the convergence (rate) and robustness of `Upcycled-FL` against statistical/device heterogeneity. Then, we adopt both output and objective perturbation to evaluate the private `Upcycled-FL`.

## 7.1 Datasets and Networks

We conduct experiments on both synthetic and real data, as detailed below. More details of each dataset are given in Appendix F.1.

---

[3]In supervised learning, the sample $d = (x, y)$ corresponds to the feature and label pair. Function $\hat{F}_i(\omega, d)$ is generalized linear model if it can be written as a function of $\omega^T x$ and $y$.

**Synthetic data.** Using the method in Li et al. (2020), we generate Syn(iid), Syn(0,0), Syn(0.5,0.5), Syn(1,1), four synthetic datasets with increasing statistical heterogeneity. We use *logistic regression* for synthetic data.

**Real data.** We adopt two real datasets: 1) FEMNIST, a federated version of EMNIST (Cohen et al., 2017). Here, A *multilayer perceptron* (MLP) consisting of two linear layers with a hidden dimension of 14x14, interconnected by ReLU activation functions, is used to learn from FEMNIST; 2) Sentiment140 (Sent140), a text sentiment analysis task on tweets (Go et al., 2009). In this context, a bidirectional LSTM with 256 hidden dimensions and 300 embedding dimensions is used to train on Sent140 dataset.

## 7.2 Experimental setup

All experiments are conducted on a server equipped with multiple NVIDIA A5000 GPUs, two AMD EPYC 7313 CPUs, and 256GB memory. The code is implemented with Python 3.8 and PyTorch 1.13.0 on Ubuntu 20.04. We employ SGD as the local optimizer, with a momentum of 0.5, and set the number of local update epochs E to 10 at each iteration $m$. Note that without privacy concerns, any classifier and loss function can be plugged into Upcycled-FL. However, if we adopt objective perturbation as privacy protection, the loss function should also satisfy assumptions in Theorem 6.3. We take the cross-entropy loss as our loss function throughout all experiments.

To simulate device heterogeneity, we randomly select a fraction of devices to train at each round, and assume there are stragglers that cannot train for full rounds; both devices and stragglers are selected by random seed to ensure they are the same for all algorithms.

**Baselines.** To evaluate the effectiveness of our strategy, we apply our Upcycled-FL method to seven representative methods in federated learning. We use the grid search to find the optimal hyperparameters for all methods to make a fair comparison. These methods include:

- FedAvg (McMahan et al., 2017): FedAvg learns the global model by averaging the client's local model.

- FedAvgM (Hsu et al., 2019): FedAvgM introduces *momentum* while averaging local models to improve convergence rates and model performance, especially in non-i.i.d. settings.

- FedProx (Li et al., 2020): FedProx adds a proximal term to the local objective function, which enables more robust convergence when data is non-i.i.d. across different clients.

- Scaffold (Karimireddy et al., 2020): Scaffold uses control variates to correct the local updates, which helps in dealing with non-i.i.d. data and accelerates convergence.

- FedDyn (Acar et al., 2021): FedDyn considers a dynamic regularization term to align the local model updates more closely with the global model.

- pFedMe (T Dinh et al., 2020): pFedMe is a personalization method to handle client heterogeneity. We set the hyperparameter $k$ in pFedMe as 5 to accelerate the training.

- FedYogi (Reddi et al., 2021): FedYogi considers the adaptive optimization for the global model aggregation.

Unless explicitly stated, the results we report are averaged outcomes over all devices. More details of experimental setup are in Appendix F.1.

## 7.3 Results

**Convergence and Heterogeneity.** Because even iterations of Upcycled-FL only involve addition/subtraction operations with no transmission overhead and almost no computational cost, we train the Upcycled version of FL algorithms with **double** iterations compared to baselines in approximately same training time in this experiment. We evaluate the convergence rate and accuracy of Upcycled-FL under different dataset and heterogeneity settings. In each iteration, 30% of devices are selected with 90% stragglers. Table 1 compares the average accuracy of different algorithms when the device heterogeneity is high (90% stragglers). The results show that all baselines can be enhanced by our methods. Notably, while FedAvg achieves good performance on Syn(iid), it is not robust on heterogeneous data, e.g. Syn(0,0), Syn(0.5,0.5), and Syn(1,1). Nonetheless, Upcycled-FedAvg makes it comparable with the regular FedProx algorithm, which shows that our strategy can also mitigate performance deterioration induced by data heterogeneity. When

Table 1: Average accuracy and standard deviation with 90% straggler on the testing dataset in the non-private setting over four runs: models are trained over synthetic data (Syn) for 80 iterations (160 for upcycled version), trained over Femnist for 150 iterations (300 for upcycled version), and trained over Sent140 for 80 iterations (160 for upcycled version). We use the grid search to find the optimal results for all methods.

| Method | Dataset | | | | | |
|---|---|---|---|---|---|---|
| | Syn(iid) | Syn(0,0) | Syn(0.5,0.5) | Syn(1,1) | FEMNIST | Sent140 |
| FedAvg | $98.06_{\pm0.07}$ | $79.28_{\pm0.61}$ | $81.58_{\pm0.43}$ | $80.40_{\pm1.28}$ | $81.38_{\pm3.54}$ | $76.11_{\pm0.11}$ |
| **Upcycled-FedAvg** | $\mathbf{98.83}_{\pm0.29}$ | $\mathbf{81.46}_{\pm0.48}$ | $\mathbf{82.89}_{\pm0.17}$ | $\mathbf{81.49}_{\pm0.53}$ | $\mathbf{82.10}_{\pm1.11}$ | $\mathbf{76.32}_{\pm0.45}$ |
| FedAvgM | $98.43_{\pm0.07}$ | $80.29_{\pm0.83}$ | $82.60_{\pm0.37}$ | $80.59_{\pm1.28}$ | $80.15_{\pm3.72}$ | $75.7_{\pm0.85}$ |
| **Upcycled-FedAvgM** | $\mathbf{98.72}_{\pm0.47}$ | $\mathbf{81.74}_{\pm0.42}$ | $\mathbf{83.13}_{\pm0.12}$ | $\mathbf{81.37}_{\pm0.82}$ | $\mathbf{81.30}_{\pm5.23}$ | $74.88_{\pm2.29}$ |
| FedProx | $96.52_{\pm0.07}$ | $80.72_{\pm0.77}$ | $81.99_{\pm0.55}$ | $81.19_{\pm0.19}$ | $79.35_{\pm0.65}$ | $73.94_{\pm0.13}$ |
| **Upcycled-FedProx** | $\mathbf{97.62}_{\pm0.32}$ | $\mathbf{80.88}_{\pm0.97}$ | $\mathbf{83.10}_{\pm0.83}$ | $\mathbf{81.94}_{\pm0.57}$ | $\mathbf{80.33}_{\pm3.43}$ | $\mathbf{74.25}_{\pm0.34}$ |
| Scaffold | $97.51_{\pm0.24}$ | $80.26_{\pm1.54}$ | $82.44_{\pm1.66}$ | $74.91_{\pm2.67}$ | $76.83_{\pm2.97}$ | $76.34_{\pm0.56}$ |
| **Upcycled-Scaffold** | $\mathbf{98.68}_{\pm0.12}$ | $\mathbf{81.10}_{\pm0.57}$ | $\mathbf{82.64}_{\pm1.39}$ | $\mathbf{76.14}_{\pm1.28}$ | $\mathbf{77.88}_{\pm5.36}$ | $\mathbf{77.34}_{\pm0.22}$ |
| FedDyn | $97.00_{\pm0.19}$ | $81.62_{\pm0.97}$ | $80.64_{\pm0.81}$ | $77.27_{\pm2.95}$ | $81.76_{\pm0.98}$ | $75.97_{\pm0.35}$ |
| **Upcycled-FedDyn** | $\mathbf{98.32}_{\pm0.08}$ | $\mathbf{82.41}_{\pm1.04}$ | $\mathbf{82.89}_{\pm0.80}$ | $\mathbf{80.03}_{\pm3.02}$ | $\mathbf{83.33}_{\pm0.71}$ | $\mathbf{76.03}_{\pm0.58}$ |
| pFedme | $96.30_{\pm0.14}$ | $89.15_{\pm0.24}$ | $89.43_{\pm0.67}$ | $93.06_{\pm0.27}$ | $71.73_{\pm4.30}$ | $72.81_{\pm0.85}$ |
| **Upcycled-pFedme** | $\mathbf{96.77}_{\pm0.12}$ | $89.08_{\pm0.22}$ | $\mathbf{89.55}_{\pm0.62}$ | $\mathbf{93.12}_{\pm0.23}$ | $\mathbf{76.66}_{\pm3.37}$ | $\mathbf{74.07}_{\pm0.74}$ |
| FedYogi | $99.30_{\pm0.35}$ | $81.20_{\pm2.64}$ | $79.49_{\pm1.30}$ | $78.95_{\pm1.98}$ | $73.53_{\pm7.73}$ | $77.02_{\pm0.07}$ |
| **Upcycled-FedYogi** | $\mathbf{99.41}_{\pm0.32}$ | $\mathbf{81.65}_{\pm2.44}$ | $\mathbf{80.84}_{\pm1.30}$ | $\mathbf{80.17}_{\pm0.84}$ | $\mathbf{75.64}_{\pm3.17}$ | $\mathbf{77.59}_{\pm0.26}$ |

data is i.i.d., FedProx with the proximal term $\frac{\mu}{2}||\omega - \overline{\omega}^t||^2$ may hurt the performance compared with FedAvg. However, the proximal term can help stabilize the algorithm and significantly improve the performance in practical settings when data is heterogeneous; these observations are consistent with (Li et al., 2020). Importantly, Upcycled-FL strategy further makes it more robust to statistical heterogeneity than regular FedProx as it could attain consistent improvements for all settings.

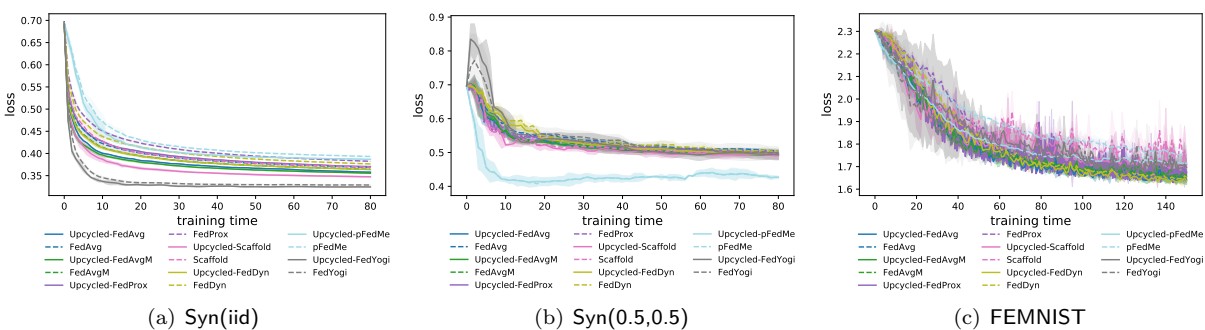

(a) Syn(iid)  (b) Syn(0.5,0.5)  (c) FEMNIST

Figure 2: Comparison on average loss and standard deviation between Upcycled-FL methods and original FL algorithms in the non-private setting under the approximate same training time. The training time refers to the time needed for a given number of iterations. Upcycled-FL does not require an update in the even iterations, allowing Upcycled-FL to train with doubled iterations.

Figure 2(a), 2(b) and 2(c) compare the convergence property on Syn(iid), Syn(0.5,0.5) and FEMNIST. Note that, by using the aggregation rule in Figure 1(b), the training time of Upcycled-FL is almost the same as the baselines without introducing extra cost. We observe that under the same training time (the number

of iterations for baselines), `Upcycled-FL` strategy brings benefits (achieving lower loss along training) for baselines in most cases. And this improvement is significant when the dataset is i.i.d. The loss trends are consistent with results in Table 1. We provide more results on the other three datasets in Appendix F.3.

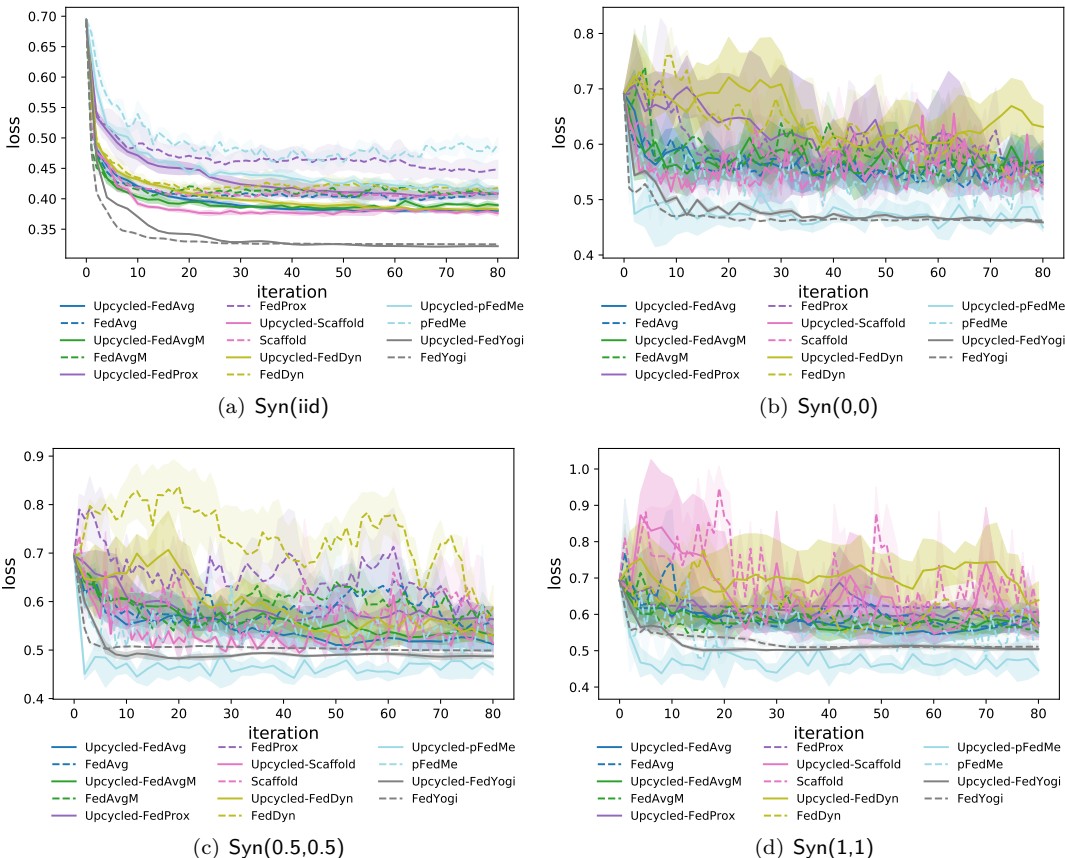

Figure 3: Comparison on average loss and standard deviation of private `Upcycled-FL` and private FL methods using **output perturbation**. The noise parameter $\sigma$ is 1.0 for all baselines, while $\sigma$ of the Upcycled version is set to 0.8. Taking the iid dataset as an example, $\bar{\epsilon} = 1.40$ for the Upcycled version, which ensures stronger privacy than the original method with $\bar{\epsilon} = 1.59$.

**Privacy-Accuracy Trade-off.** We next inspect the accuracy-privacy trade-off of private `Upcycled-FL` and compare it with private baselines. Although we adopt both objective and output perturbation to achieve differential privacy, other techniques can also be used. For each parameter setting, we still conduct a grid search and perform 4 independent runs of experiments. To precisely quantify privacy, no straggler is considered in this experiment. We reported the results using output perturbation and objective perturbation. Figure 3 and 4 demonstrate the performance of private `Upcylced-FL` and private baselines on synthetic data using two types of perturbation respectively. We also report the results on real data in Appendix F.4.

Here we carefully set the perturbation strength of each algorithm such that the privacy loss $\epsilon$ for private `Upcycled-FL` is strictly less than the original methods. As expected, private `Upcycled-FL` is more stable than baselines and attains a lower loss value under smaller $\epsilon$. This is because the private `Upcycled-FL` with less information leakage requires much less perturbation to attain the same privacy guarantee as FL methods under privacy constraints. We also observe that in general `Upcycled-FL` can be used to augment all baseline methods with or without client local heterogeneity.

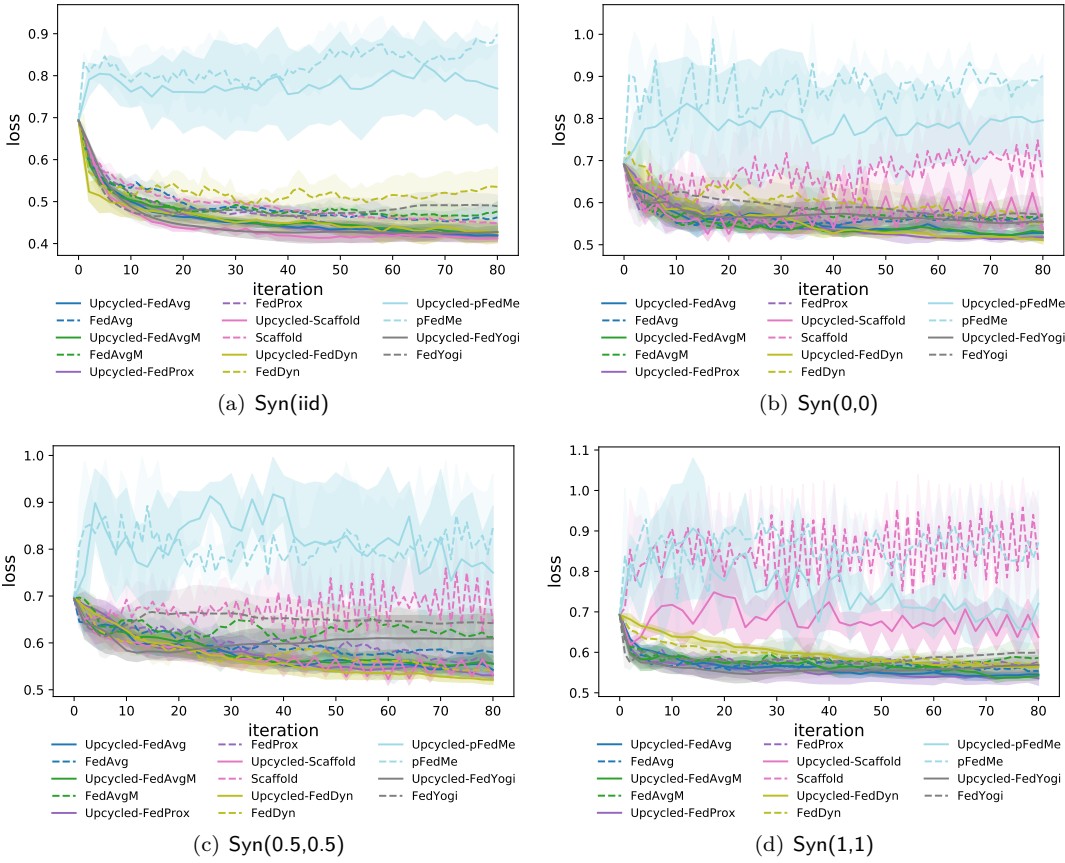

Figure 4: Comparison on average loss and standard deviation of private `Upcycled-FL` and private FL methods using **objective perturbation**. Under objective perturbation, noise parameter $\alpha$ is 10 for all baselines, while $\alpha$ of the Upcycled version is set to 20 to ensure stronger privacy than the original versions. Taking the iid dataset as an example, $\bar{\epsilon}$ associated with these noise parameters is 7.36 for `FedProx` and 7.25 for `Upcycled-FedProx` (when $\mu = 0.5$).

## 8 Conclusion

This paper proposes `Upcycled-FL`, a novel plug-in federated learning strategy under which information leakage and computation costs can be reduced significantly. We theoretically examined the convergence (rate) of `Upcycled-FedProx`, a special case when `Upcycled-FL` strategy is applied to a well-known FL algorithm named `FedProx`. Extensive experiments on synthetic and read data further show that `Upcycled-FL` can be combined with common FL algorithms and enhance their robustness on heterogeneous data while attaining much better privacy-accuracy trade-off under common differential privacy mechanisms.

## Acknowledgments

This material is based upon work supported by the U.S. National Science Foundation under awards IIS-2040800, IIS-2112471, IIS-2202699, IIS-2301599, and CMMI-2301601, by grants from the Ohio State University's Translational Data Analytics Institute and College of Engineering Strategic Research Initiative.

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

## A  Notation Table

| | |
|---|---|
| $\mathcal{I}$ | set of agents |
| $\mathcal{D}_i$ | dataset of agent $i$ |
| $p_i$ | size of agent $i$'s data as a fraction of total data samples |
| $\omega_i^t$ | agent $i$'s local model parameter at time $t$ |
| $\overline{\omega}^t$ | aggregated model at central server at $t$ |
| $\widehat{\omega}_i^t$ | differentially private version of $\omega_i^t$ |
| $F_i$ | local objective function of agent $i$ |
| $f$ | overall objective function |
| $n_i^t$ | random noise added to agent $i$ at time $t$ |
| $\mu$ | hyper-parameter for proximal term in `FedProx` and `Upcycled FedProx` |
| $\lambda_m$ | hyper-parameter for first-order approximation at even iteration $2m$ in `Upcycled-FL` |
| $\tau$ | the clipping threshold for output perturbation |

## B  Lemmas

**Lemma B.1.** *Define $\widetilde{\omega}^t = \mathbb{E}_i(\omega_i^t)$. Suppose conditions in Theorem 5.6 hold, then the following holds $\forall m$:*

$$
\begin{aligned}
f(\widetilde{\omega}^{2m+1}) \;\leq\; & f(\overline{\omega}^{2m-1}) - \widehat{C}_1\Big(L, B, \frac{1}{\mu}, \frac{1}{\rho}\Big)||\nabla f(\overline{\omega}^{2m-1})||^2 \\
& + \widehat{C}_2\Big(L, B, \frac{1}{\mu}, \frac{1}{\rho}, \frac{\mu}{\mu+\lambda_m}\Big)||\nabla f(\overline{\omega}^{2m-1})|| \cdot ||\overline{\omega}^{2m-1} - \overline{\omega}^{2m-2}|| \\
& + \widehat{C}_3\Big(L, B, \frac{1}{\mu}, \frac{1}{\rho}, \frac{\mu}{\mu+\lambda_m}\Big)||\overline{\omega}^{2m-1} - \overline{\omega}^{2m-2}||^2
\end{aligned}
$$

*where coefficients satisfy*

$$
\begin{aligned}
\widehat{C}_1\Big(L, B, \frac{1}{\mu}, \frac{1}{\rho}\Big) &= \frac{1}{\mu} - \frac{LB}{\mu^2\rho} - \frac{LB^2}{2\rho^2} \\
\widehat{C}_2\Big(L, B, \frac{1}{\mu}, \frac{1}{\rho}, \frac{\mu}{\mu+\lambda_m}\Big) &= \Big(\frac{L^2}{\mu^2\rho} + \frac{L+\mu}{\mu^2} + \frac{L(L+\rho)B}{\rho^2}\Big)\frac{\mu}{\mu+\lambda_m} \\
\widehat{C}_3\Big(L, B, \frac{1}{\mu}, \frac{1}{\rho}, \frac{\mu}{\mu+\lambda_m}\Big) &= \frac{L(L+\rho)^2}{2\rho^2}\frac{\mu^2}{(\mu+\lambda_m)^2}
\end{aligned}
$$

**Lemma B.2.** *Let $\mathcal{S}_m$ be the set of $K$ randomly selected local devices got updated at iterations $2m-1$ and $2m$, and $\mathbb{E}_{\mathcal{S}_m}[\cdot]$ be the expectation with respect to the choice of devices. Then we have*

$$
\begin{aligned}
\mathbb{E}_{\mathcal{S}_m}[f(\overline{\omega}^{2m+1})] \;\leq\; & f(\widetilde{\omega}^{2m+1}) + \widetilde{C}_1\Big(B, L, \frac{1}{K}, \frac{1}{\rho}\Big)||\nabla f(\overline{\omega}^{2m-1})||^2 \\
& + \widetilde{C}_2\Big(B, L, \frac{1}{K}, \frac{1}{\rho}, \frac{\mu}{\mu+\lambda_m}\Big)||\nabla f(\overline{\omega}^{2m-1})|| \cdot ||\overline{\omega}^{2m-1} - \overline{\omega}^{2m-2}|| \\
& + \widetilde{C}_3\Big(B, L, \frac{1}{K}, \frac{1}{\rho}, \frac{\mu}{\mu+\lambda_m}\Big)||\overline{\omega}^{2m-1} - \overline{\omega}^{2m-2}||^2
\end{aligned}
$$

*where coefficients satisfy*

$$
\begin{aligned}
\widetilde{C}_1\Big(B, L, \frac{1}{K}, \frac{1}{\rho}\Big) &= \frac{2B^2}{K\rho^2} + \frac{2LB+\rho}{\rho}\sqrt{\frac{2}{K}}\frac{B}{\rho} \\
\widetilde{C}_2\Big(B, L, \frac{1}{K}, \frac{1}{\rho}, \frac{\mu}{\mu+\lambda_m}\Big) &= \Big(\frac{4LB}{K\rho^2} + \frac{2LB+\rho}{\rho}\sqrt{\frac{2}{K}}\frac{L}{\rho} + 2L\frac{L+\rho}{\rho}\sqrt{\frac{2}{K}}\frac{B}{\rho}\Big) \cdot \frac{\mu}{\mu+\lambda_m} \\
\widetilde{C}_3\Big(B, L, \frac{1}{K}, \frac{1}{\rho}, \frac{\mu}{\mu+\lambda_m}\Big) &= \Big(\frac{2}{K}\frac{L^2}{\rho^2} + 2L\frac{L+\rho}{\rho}\sqrt{\frac{2}{K}}\frac{L}{\rho}\Big) \cdot \Big(\frac{\mu}{\mu+\lambda_m}\Big)^2
\end{aligned}
$$

## C   Proofs

**Proof of Lemma B.1**

*Proof.* Since local functions $F_i$ are $L$-Lipschitz smooth, at iteration $2m-1$, we have

$$f(\widetilde{\omega}^{2m+1})$$

$$\leq f(\overline{\omega}^{2m-1}) + \langle \nabla f(\overline{\omega}^{2m-1}), \widetilde{\omega}^{2m+1} - \overline{\omega}^{2m-1} \rangle + \frac{L}{2}||\widetilde{\omega}^{2m+1} - \overline{\omega}^{2m-1}||^2$$

$$= f(\overline{\omega}^{2m-1}) + \langle \nabla f(\overline{\omega}^{2m-1}), -\frac{1}{\mu}\nabla f(\overline{\omega}^{2m-1}) + \Phi^{2m+1} \rangle + \frac{L}{2}||\widetilde{\omega}^{2m+1} - \overline{\omega}^{2m-1}||^2$$

$$\leq f(\overline{\omega}^{2m-1}) - \frac{1}{\mu}||\nabla f(\overline{\omega}^{2m-1})||^2 + \frac{1}{\mu}||\nabla f(\overline{\omega}^{2m-1})|| \cdot ||\Phi^{2m+1}|| + \frac{L}{2}||\widetilde{\omega}^{2m+1} - \overline{\omega}^{2m-1}||^2$$

where

$$\Phi^{2m+1} = \frac{1}{\mu}\nabla f(\overline{\omega}^{2m-1}) + \widetilde{\omega}^{2m+1} - \overline{\omega}^{2m-1} = \mathbb{E}_i\Big[\frac{1}{\mu}\nabla F_i(\overline{\omega}^{2m-1}) + \omega_i^{2m+1} - \overline{\omega}^{2m-1}\Big] \tag{7}$$

By first-order condition, the following holds at $(2m+1)$-th iteration:

$$\omega_i^{2m+1} - \overline{\omega}^{2m-1} = -\frac{1}{\mu}\nabla F_i(\omega_i^{2m+1}) + \overline{\omega}^{2m} - \overline{\omega}^{2m-1}$$

Plug into Eqn. (7), we have

$$\Phi^{2m+1} = \mathbb{E}_i\Big[\frac{1}{\mu}\Big(\nabla F_i(\overline{\omega}^{2m-1}) - \nabla F_i(\omega_i^{2m+1})\Big) + \overline{\omega}^{2m} - \overline{\omega}^{2m-1}\Big]$$

By $L$-Lipschitz smoothness and Jensen's inequality, we have

$$||\Phi^{2m+1}|| \leq \mathbb{E}_i\Big[\frac{1}{\mu}||\nabla F_i(\overline{\omega}^{2m-1}) - \nabla F_i(\omega_i^{2m+1})||\Big] + ||\overline{\omega}^{2m} - \overline{\omega}^{2m-1}|| \tag{8}$$

$$\leq \mathbb{E}_i\Big[\frac{L}{\mu}||\overline{\omega}^{2m-1} - \omega_i^{2m+1}||\Big] + ||\overline{\omega}^{2m} - \overline{\omega}^{2m-1}||$$

$$\leq \mathbb{E}_i\Big[\frac{L}{\mu}||\omega_i^{2m+1} - \overline{\omega}^{2m}||\Big] + \frac{L+\mu}{\mu}||\overline{\omega}^{2m} - \overline{\omega}^{2m-1}||$$

Since $h_i$ are $\rho$-strongly convex, $F_i$ is $L$-Lipschitz smooth, and $\omega_i^{2m+1} = \arg\min_\omega h_i(\omega; \overline{\omega}^{2m})$ we have

$$||\omega_i^{2m+1} - \overline{\omega}^{2m}|| \leq \frac{1}{\rho}||\nabla h_i(\omega_i^{2m+1}; \overline{\omega}^{2m}) - \nabla h_i(\overline{\omega}^{2m}; \overline{\omega}^{2m})|| = \frac{1}{\rho}||0 - \nabla F_i(\overline{\omega}^{2m})||$$

$$\leq \frac{L}{\rho}||\overline{\omega}^{2m} - \overline{\omega}^{2m-1}|| + \frac{1}{\rho}||\nabla F_i(\overline{\omega}^{2m-1})|| \tag{9}$$

Plug in Eqn. (8),

$$||\Phi^{2m+1}|| \leq \frac{L}{\mu\rho}\mathbb{E}_i[||\nabla F_i(\overline{\omega}^{2m-1})||] + \Big(\frac{L^2}{\mu\rho} + \frac{L+\mu}{\mu}\Big)||\overline{\omega}^{2m} - \overline{\omega}^{2m-1}||$$

Consider the following term

$$||\widetilde{\omega}^{2m+1} - \overline{\omega}^{2m-1}|| = ||\mathbb{E}_i[\omega_i^{2m+1}] - \overline{\omega}^{2m-1}|| \leq \mathbb{E}_i[||\omega_i^{2m+1} - \overline{\omega}^{2m-1}||] \tag{10}$$

$$\leq \mathbb{E}_i[||\omega_i^{2m+1} - \overline{\omega}^{2m}|| + ||\overline{\omega}^{2m} - \overline{\omega}^{2m-1}||]$$

$$\leq \frac{L+\rho}{\rho}||\overline{\omega}^{2m} - \overline{\omega}^{2m-1}|| + \frac{1}{\rho}\mathbb{E}_i\Big[||\nabla F_i(\overline{\omega}^{2m-1})||\Big]$$

Because $F_i$ is $B$-dissimilar, we have

$$\mathbb{E}_i\Big[||\nabla F_i(\overline{\omega}^{2m-1})||\Big] \le B||\nabla f(\overline{\omega}^{2m-1})||$$

Therefore,

$$f(\widetilde{\omega}^{2m+1})$$
$$\le f(\overline{\omega}^{2m-1}) - \frac{1}{\mu}||\nabla f(\overline{\omega}^{2m-1})||^2 + \frac{1}{\mu}||\nabla f(\overline{\omega}^{2m-1})|| \cdot ||\Phi^{2m+1}|| + \frac{L}{2}||\widetilde{\omega}^{2m+1} - \overline{\omega}^{2m-1}||^2$$
$$\le f(\overline{\omega}^{2m-1}) - \Big(\frac{1}{\mu} - \frac{LB}{\mu^2\rho} - \frac{LB^2}{2\rho^2}\Big)||\nabla f(\overline{\omega}^{2m-1})||^2$$
$$+ \Big(\frac{L^2}{\mu^2\rho} + \frac{L+\mu}{\mu^2} + \frac{L(L+\rho)B}{\rho^2}\Big)||\nabla f(\overline{\omega}^{2m-1})|| \cdot ||\overline{\omega}^{2m} - \overline{\omega}^{2m-1}||$$
$$+ \frac{L(L+\rho)^2}{2\rho^2}||\overline{\omega}^{2m} - \overline{\omega}^{2m-1}||^2$$

After applying first-order approximation in even iterations, we have

$$\overline{\omega}^{2m} - \overline{\omega}^{2m-1} = \frac{\mu}{\mu+\lambda_m}(\overline{\omega}^{2m-1} - \overline{\omega}^{2m-2})$$

Therefore,

$$
\begin{aligned}
f(\widetilde{\omega}^{2m+1}) \le\ & f(\overline{\omega}^{2m-1}) - \Big(\frac{1}{\mu} - \frac{LB}{\mu^2\rho} - \frac{LB^2}{2\rho^2}\Big)||\nabla f(\overline{\omega}^{2m-1})||^2 \\
& + \Big(\frac{L^2}{\mu^2\rho} + \frac{L+\mu}{\mu^2} + \frac{L(L+\rho)B}{\rho^2}\Big)\frac{\mu}{\mu+\lambda_m}||\nabla f(\overline{\omega}^{2m-1})|| \cdot ||\overline{\omega}^{2m-1} - \overline{\omega}^{2m-2}|| \\
& + \frac{L(L+\rho)^2}{2\rho^2}\frac{\mu^2}{(\mu+\lambda_m)^2}||\overline{\omega}^{2m-1} - \overline{\omega}^{2m-2}||^2
\end{aligned}
$$

The Lemma B.1 is proved. $\qquad\square$

**Proof of Lemma B.2**

*Proof.* Because local function $F_i$ is $L$-Lipschitz smooth, $f$ is local Lipschitz continuous.

$$f(\overline{\omega}^{2m+1}) \le f(\widetilde{\omega}^{2m+1}) + L_0||\overline{\omega}^{2m+1} - \widetilde{\omega}^{2m+1}||$$

where $L_0$ is the local Lipschitz continuouty constant. Moreover, we have

$$L_0 \le ||\nabla f(\overline{\omega}^{2m-1})|| + L(||\widetilde{\omega}^{2m+1} - \overline{\omega}^{2m-1}|| + ||\overline{\omega}^{2m+1} - \overline{\omega}^{2m-1}||)$$

Therefore,

$$
\begin{aligned}
\mathbb{E}_{\mathcal{S}_m}[f(\overline{\omega}^{2m+1})] \le\ & f(\widetilde{\omega}^{2m+1}) \\
& + \mathbb{E}_{\mathcal{S}_m}\Big[\Big(||\nabla f(\overline{\omega}^{2m-1})|| + L(||\widetilde{\omega}^{2m+1} - \overline{\omega}^{2m-1}|| + ||\overline{\omega}^{2m+1} - \overline{\omega}^{2m-1}||)\Big)||\overline{\omega}^{2m+1} - \widetilde{\omega}^{2m+1}||\Big] \\
=\ & f(\widetilde{\omega}^{2m+1}) + \Big(||\nabla f(\overline{\omega}^{2m-1})|| + L||\widetilde{\omega}^{2m+1} - \overline{\omega}^{2m-1}||\Big) \cdot \mathbb{E}_{\mathcal{S}_m}[||\overline{\omega}^{2m+1} - \widetilde{\omega}^{2m+1}||] \\
& + L\mathbb{E}_{\mathcal{S}_m}\Big[||\overline{\omega}^{2m+1} - \overline{\omega}^{2m-1}|| \cdot ||\overline{\omega}^{2m+1} - \widetilde{\omega}^{2m+1}||\Big] \\
\le\ & f(\widetilde{\omega}^{2m+1}) + \Big(||\nabla f(\overline{\omega}^{2m-1})|| + 2L||\widetilde{\omega}^{2m+1} - \overline{\omega}^{2m-1}||\Big) \cdot \mathbb{E}_{\mathcal{S}_m}[||\overline{\omega}^{2m+1} - \widetilde{\omega}^{2m+1}||] \\
& + L\mathbb{E}_{\mathcal{S}_m}\Big[||\overline{\omega}^{2m+1} - \widetilde{\omega}^{2m+1}||^2\Big]
\end{aligned}
$$

When $K$ devices are randomly selected, by Eqn. (9), we have

$$
\begin{aligned}
&\mathbb{E}_{\mathcal{S}_m}\Big[||\overline{\omega}^{2m+1} - \widetilde{\omega}^{2m+1}||^2\Big] \\
&\leq \frac{1}{K}\mathbb{E}_i\Big[||\omega_i^{2m+1} - \widetilde{\omega}^{2m+1}||^2\Big] \leq \frac{2}{K}\mathbb{E}_i\Big[||\omega_i^{2m+1} - \overline{\omega}^{2m}||^2\Big] \\
&\leq \frac{2}{K}\mathbb{E}_i\Big[\frac{L^2}{\rho^2}||\overline{\omega}^{2m} - \overline{\omega}^{2m-1}||^2 + \frac{1}{\rho^2}||\nabla F_i(\overline{\omega}^{2m-1})||^2 + \frac{2L}{\rho^2}||\overline{\omega}^{2m} - \overline{\omega}^{2m-1}|| \cdot ||\nabla F_i(\overline{\omega}^{2m-1})||\Big] \\
&\leq \frac{2}{K}\frac{L^2}{\rho^2}||\overline{\omega}^{2m} - \overline{\omega}^{2m-1}||^2 + \frac{2B^2}{K\rho^2}||\nabla f(\overline{\omega}^{2m-1})||^2 + \frac{4LB}{K\rho^2}||\overline{\omega}^{2m} - \overline{\omega}^{2m-1}|| \cdot ||\nabla f(\overline{\omega}^{2m-1})|| \\
&= \frac{2}{K}\Big(\frac{L}{\rho}||\overline{\omega}^{2m} - \overline{\omega}^{2m-1}|| + \frac{B}{\rho}||\nabla f(\overline{\omega}^{2m-1})||\Big)^2
\end{aligned}
$$

By Jensen's inequality,

$$
\begin{aligned}
\mathbb{E}_{\mathcal{S}_m}\Big[||\overline{\omega}^{2m+1} - \widetilde{\omega}^{2m+1}||\Big] &\leq \sqrt{\mathbb{E}_{\mathcal{S}_m}\Big[||\overline{\omega}^{2m+1} - \widetilde{\omega}^{2m+1}||^2\Big]} \\
&= \sqrt{\frac{2}{K}}\Big(\frac{L}{\rho}||\overline{\omega}^{2m} - \overline{\omega}^{2m-1}|| + \frac{B}{\rho}||\nabla f(\overline{\omega}^{2m-1})||\Big)
\end{aligned}
$$

By Eqn. (10),

$$
||\nabla f(\overline{\omega}^{2m-1})|| + 2L||\widetilde{\omega}^{2m+1} - \overline{\omega}^{2m-1}|| \leq 2L\frac{L+\rho}{\rho}||\overline{\omega}^{2m} - \overline{\omega}^{2m-1}|| + \frac{2LB+\rho}{\rho}||\nabla f(\overline{\omega}^{2m-1})||
$$

Re-organize, we have

$$
\begin{aligned}
\mathbb{E}_{\mathcal{S}_m}[f(\overline{\omega}^{2m+1})] \leq &f(\widetilde{\omega}^{2m+1}) + \frac{2}{K}\frac{L^2}{\rho^2}||\overline{\omega}^{2m} - \overline{\omega}^{2m-1}||^2 + \frac{2B^2}{K\rho^2}||\nabla f(\overline{\omega}^{2m-1})||^2 \\
&+ \frac{4LB}{K\rho^2}||\overline{\omega}^{2m} - \overline{\omega}^{2m-1}|| \cdot ||\nabla f(\overline{\omega}^{2m-1})|| \\
&+ \Big(2L\frac{L+\rho}{\rho}||\overline{\omega}^{2m} - \overline{\omega}^{2m-1}|| + \frac{2LB+\rho}{\rho}||\nabla f(\overline{\omega}^{2m-1})||\Big)\sqrt{\frac{2}{K}}\frac{L}{\rho}||\overline{\omega}^{2m} - \overline{\omega}^{2m-1}|| \\
&+ \Big(2L\frac{L+\rho}{\rho}||\overline{\omega}^{2m} - \overline{\omega}^{2m-1}|| + \frac{2LB+\rho}{\rho}||\nabla f(\overline{\omega}^{2m-1})||\Big)\sqrt{\frac{2}{K}}\frac{B}{\rho}||\nabla f(\overline{\omega}^{2m-1})|| \\
= &f(\widetilde{\omega}^{2m+1}) + \Big(\frac{2}{K}\frac{L^2}{\rho^2} + 2L\frac{L+\rho}{\rho}\sqrt{\frac{2}{K}}\frac{L}{\rho}\Big) \cdot \Big(\frac{\mu}{\mu+\lambda_m}\Big)^2||\overline{\omega}^{2m-1} - \overline{\omega}^{2m-2}||^2 \\
&+ \Big(\frac{4LB}{K\rho^2} + \frac{2LB+\rho}{\rho}\sqrt{\frac{2}{K}}\frac{L}{\rho} + 2L\frac{L+\rho}{\rho}\sqrt{\frac{2}{K}}\frac{B}{\rho}\Big) \cdot \frac{\mu}{\mu+\lambda_m}||\overline{\omega}^{2m-1} - \overline{\omega}^{2m-2}|| \cdot ||\nabla f(\overline{\omega}^{2m-1})|| \\
&+ \Big(\frac{2B^2}{K\rho^2} + \frac{2LB+\rho}{\rho}\sqrt{\frac{2}{K}}\frac{B}{\rho}\Big)||\nabla f(\overline{\omega}^{2m-1})||^2
\end{aligned}
$$

Lemma B.2 is proved. $\qquad\square$

**Proof of Lemma 5.5**

*Proof.* Lemma 5.5 can be proved by combing Lemmas B.1 and B.2, where

$$
\begin{aligned}
\boldsymbol{C_1} \;:=\; & C_1\Big(L, B, \frac{1}{\mu}, \frac{1}{\rho}, \frac{1}{K}\Big) = \widehat{C}_1\Big(L, B, \frac{1}{\mu}, \frac{1}{\rho}\Big) - \widetilde{C}_1\Big(L, B, \frac{1}{K}, \frac{1}{\rho}\Big) \\
=\; & \frac{1}{\mu} - \frac{LB}{\mu^2\rho} - \frac{LB^2}{2\rho^2} - \frac{2B^2}{K\rho^2} - \frac{2LB+\rho}{\rho}\sqrt{\frac{2}{K}}\frac{B}{\rho}
\end{aligned}
\tag{11}
$$

$$
\begin{aligned}
\boldsymbol{C_2} \;:=\; & C_2\Big(L, B, \frac{1}{\mu}, \frac{1}{\rho}, \frac{1}{K}, \frac{\mu}{\mu+\lambda_m}\Big) = \widehat{C}_2\Big(L, B, \frac{1}{\mu}, \frac{1}{\rho}, \frac{\mu}{\mu+\lambda_m}\Big) + \widetilde{C}_2\Big(L, B, \frac{1}{K}, \frac{1}{\rho}, \frac{\mu}{\mu+\lambda_m}\Big) \\
=\; & \Big(\frac{L^2}{\mu^2\rho} + \frac{L+\mu}{\mu^2} + \frac{L(L+\rho)B}{\rho^2} + \frac{4LB}{K\rho^2} + \frac{(4L^2B+\rho L(1+2B))}{\rho^2}\sqrt{\frac{2}{K}}\Big) \cdot \frac{\mu}{\mu+\lambda_m}
\end{aligned}
\tag{12}
$$

$$
\begin{aligned}
\boldsymbol{C_3} \;:=\; & C_3\Big(L, \frac{1}{\mu}, \frac{1}{\rho}, \frac{1}{K}, \frac{\mu}{\mu+\lambda_m}\Big) = \widehat{C}_3\Big(L, \frac{1}{\mu}, \frac{1}{\rho}, \frac{\mu}{\mu+\lambda_m}\Big) + \widetilde{C}_3\Big(L, \frac{1}{K}, \frac{1}{\rho}, \frac{\mu}{\mu+\lambda_m}\Big) \\
=\; & \Big(\frac{L(L+\rho)^2}{2\rho^2} + \frac{2}{K}\frac{L^2}{\rho^2} + 2L\frac{L+\rho}{\rho}\sqrt{\frac{2}{K}}\frac{L}{\rho}\Big) \cdot \Big(\frac{\mu}{\mu+\lambda_m}\Big)^2
\end{aligned}
\tag{13}
$$

$\square$

# D   Proof of Theorem 5.6

This can be proved directed using Lemma 5.5 by averaging over all $M$ odd iterations.

# E   Proof of Theorem 6.2

WLOG, consider the case when local device got updated in every iteration and the algorithm runs over $2M$ iterations in total. We will use the uppercase letters $X$ and lowercase letters $x$ to denote random variables and the corresponding realizations, and use $P_X(\cdot)$ to denote its probability distribution. To simplify the notations, we will drop the index $i$ as we are only concerned with one agent.

According to Abadi et al. (2016), for a mechanism $\mathcal{M}$ outputs $o$, with inputs $d$ and $\hat{d}$, let a random variable $c(o; \mathcal{M}, d, \hat{d}) = \log\frac{\Pr(\mathcal{M}(d)=o)}{\Pr(\mathcal{M}(\hat{d})=o)}$ denote the privacy loss at $o$, and

$$
\alpha_{\mathcal{M}}(\lambda) = \max_{d, \hat{d}} \log \mathbb{E}_{o \sim \mathcal{M}(d)}\{\exp(\lambda c(o; \mathcal{M}, d, \hat{d}))\}
$$

For two neighboring datasets $\mathcal{D}$ and $\mathcal{D}'$ of agent $i$, by Lemma 6.1, the total privacy loss is only contributed by odd iterations. Thus, for any sequence of private (clipped) models $\widehat{\omega}^t$ generated by mechanisms $\{\mathcal{M}^m\}_{m=1}^M$ over $2M$ iterations, there is:

$$
\begin{aligned}
c(\widehat{\omega}^{0:2M}; \{\mathcal{M}^m\}_{m=1}^M, \mathcal{D}, \mathcal{D}') \;=\; & \log \frac{P_{\widehat{\Omega}^{0:2M}}(\widehat{\omega}^{0:2M}|\mathcal{D})}{P_{\widehat{\Omega}^{0:2M}}(\widehat{\omega}^{0:2M}|\mathcal{D}')} \\
=\; & \sum_{m=0}^M \log \frac{P_{\widehat{\Omega}^{2m+1}}(\widehat{\omega}^{2m+1}|\mathcal{D}, \widehat{\omega}^{0:2m})}{P_{\widehat{\Omega}^{2m+1}}(\widehat{\omega}^{2m+1}|\mathcal{D}', \widehat{\omega}^{0:2m})} + \log \frac{P_{\widehat{\Omega}^0}(\widehat{\omega}^0|\mathcal{D})}{P_{\widehat{\Omega}^0}(\widehat{\omega}^0|\mathcal{D}')} \\
=\; & \sum_{m=0}^M c(\widehat{\omega}^{2m+1}; \mathcal{M}^m, \widehat{\omega}^{0:2m}, \mathcal{D}, \mathcal{D}')
\end{aligned}
$$

where $\widehat{\omega}^{0:t} = \{\widehat{\omega}^\tau\}_{\tau=0}^t$ and $\widehat{\Omega}^t$ is random variable whose realization is $\widehat{\omega}^t$. Since $\widehat{\omega}^0$ is randomly generated, which is independent of dataset, we have $P_{\widehat{\Omega}^0}(\widehat{\omega}^0|\mathcal{D}) = P_{\widehat{\Omega}^0}(\widehat{\omega}^0|\mathcal{D}')$. Moreover, the following holds for any $\lambda$:

$$\log \mathbb{E}_{\widehat{\omega}^{0:2M}}\{\exp(\lambda c(\widehat{\omega}^{0:2M}; \{\mathcal{M}^m\}_{m=1}^M, \mathcal{D}, \mathcal{D}'))\}$$

$$= \log \mathbb{E}_{\widehat{\omega}^{0:2M}}\{\exp(\lambda \sum_{m=0}^M c(\widehat{\omega}^{2m+1}; \mathcal{M}^m, \widehat{\omega}^{0:2m}, \mathcal{D}, \mathcal{D}')\}$$

$$= \sum_{m=0}^M \log \mathbb{E}_{\widehat{\omega}^{2m+1}}\{\exp(\lambda c(\widehat{\omega}^{2m+1}; \mathcal{M}^m, \widehat{\omega}^{0:2m}, \mathcal{D}, \mathcal{D}')\} \tag{14}$$

Therefore, $\alpha_{\{\mathcal{M}^m\}_{m=1}^M}(\lambda) \le \sum_{m=1}^M \alpha_{\mathcal{M}^m}(\lambda)$ also holds. First bound each individual $\alpha_{\mathcal{M}^m}(\lambda)$.

Consider two neighboring datasets $\mathcal{D}$ and $\mathcal{D}'$. Private (clipped) model $\widehat{\omega}^{2m+1}$ is generated by mechanism $\mathcal{M}^m(D) = \xi(\omega^{2m+1}) + N = \frac{1}{|D|}\sum_{d \in D} \eta(d) + N$ with function $||\eta(\cdot)||_2 \le \tau$ and Gaussian noise $N \sim \mathcal{N}(0, \sigma^2 \mathbf{I})$. Without loss of generality, let $\mathcal{D}' = \mathcal{D} \cup \{d_n\}$, $f(d_n) = \pm\tau\mathbf{e}_1$ and $\sum_{d \in \mathcal{D}} \eta(d) = \mathbf{0}$. Then $\mathcal{M}^m(\mathcal{D})$ and $\mathcal{M}^m(\mathcal{D}')$ are distributed identically except for the first coordinate and the problem can be reduced to one-dimensional problem.

$$c(\widehat{\omega}^{2m+1}; \mathcal{M}^m, \widehat{\omega}^{0:2m}, \mathcal{D}, \mathcal{D}') = \log \frac{P_{\widehat{\Omega}^{2m+1}}(\widehat{\omega}^{2m+1}|\mathcal{D}, \widehat{\omega}^{0:2m})}{P_{\widehat{\Omega}^{2m+1}}(\widehat{\omega}^{2m+1}|\mathcal{D}', \widehat{\omega}^{0:2m})}$$

$$= \log \frac{P_N(n)}{P_N(n \pm \tau)}$$

$$\le \frac{\tau}{2|D|\sigma^2}(2|n| + \tau).$$

where $n + \frac{1}{|D|}\sum_{d \in \mathcal{D}} \eta(d) = \widehat{\omega}^{2m+1}$. Therefore,

$$\alpha_{\mathcal{M}^m}(\lambda) = \log \mathbb{E}_{N \sim \mathcal{N}(0,\sigma^2)}\{\exp(\lambda \frac{\tau}{2|D|\sigma^2}(2N + \tau))\}$$

$$= \log \int_{-\infty}^{\infty} \frac{1}{\sqrt{2\pi}\sigma} \exp(-\frac{1}{2\sigma^2}(n - \lambda\frac{\tau}{|D|})^2) \cdot \exp(\frac{\tau^2}{2|D|^2\sigma^2}(\lambda^2 + \lambda))dn$$

$$= \frac{\tau^2\lambda(\lambda+1)}{2|D|^2\sigma^2}.$$

$$\alpha_{\{\mathcal{M}^m\}_{m=1}^M}(\lambda) \le \sum_{m=1}^M \alpha_{\mathcal{M}^m}(\lambda) = \frac{M\tau^2\lambda(\lambda+1)}{2|D|^2\sigma^2}$$

Use the tail bound [Theorem 2, Abadi et al. (2016)], for any $\varepsilon \ge \frac{M\tau^2}{2|D|^2\sigma^2}$, the algorithm is $(\varepsilon, \delta)$-differentially private for

$$\delta = \min_{\lambda:\lambda \ge 0} h(\lambda) = \min_{\lambda:\lambda \ge 0} \exp\left(\frac{M\tau^2\lambda(\lambda+1)}{2|D|^2\sigma^2} - \lambda\varepsilon\right)$$

To find $\lambda^* = \underset{\lambda:\lambda \ge 0}{\arg\min}\, h(\lambda)$, take derivative of $h(\lambda)$ and assign 0 gives the solution $\bar{\lambda} = \frac{\varepsilon|D|^2\sigma^2}{M\tau^2} - \frac{1}{2} \ge 0$, and $h''(\bar{\lambda}) > 0$, implies $\lambda^* = \bar{\lambda}$. Plug into (15) gives:

$$\delta = \exp\left(\left(\frac{M\tau^2}{4|D|^2\sigma^2} - \frac{\varepsilon}{2}\right)\left(\frac{\varepsilon|D|^2\sigma^2}{M\tau^2} - \frac{1}{2}\right)\right) \tag{15}$$

Similarly, for any $\delta \in [0, 1]$, the algorithm is $(\varepsilon, \delta)$-differentially private for

$$\varepsilon = \min_{\lambda:\lambda \ge 0} h_1(\lambda) = \min_{\lambda:\lambda \ge 0} \frac{M\tau^2(\lambda+1)}{2|D|^2\sigma^2} + \frac{1}{\lambda}\log\left(\frac{1}{\delta}\right) = 2\sqrt{\frac{M\tau^2}{2|D|^2\sigma^2}\log(\frac{1}{\delta})} + \frac{M\tau^2}{2|D|^2\sigma^2}$$

**Proof of Theorem 6.3**

*Proof.* WLOG, consider the case when local device got updated in every iteration and the algorithm runs over $2M$ iteration in total.

We will use the uppercase letters $X$ and lowercase letters $x$ to denote random variables and the corresponding realizations, and use $P_X(\cdot)$ to denote its probability distribution. To simplify the notations, we will drop the index $i$ as we are only concerned with one agent, and use $\omega^t$ to denote private output $\widehat{\omega}^t$.

For two neighboring datasets $\mathcal{D}$ and $\mathcal{D}'$ of agent $i$, by Lemma 6.1, the total privacy loss is only contributed by odd iterations. Thus, the ratio of joint probabilities (privacy loss) is given by:

$$\frac{P_{\Omega^{0:2M}}(\omega^{0:2M}|\mathcal{D})}{P_{\Omega^{0:2M}}(\omega^{0:2M}|\mathcal{D}')} = \frac{P_{\Omega^0}(\omega^0|\mathcal{D})}{P_{\Omega^0}(\omega^0|\mathcal{D}')} \cdot \prod_{m=0}^{M} \frac{P_{\Omega^{2m+1}}(\omega^{2m+1}|\omega^{0:2m},\mathcal{D})}{P_{\Omega^{2m+1}}(\omega^{2m+1}|\omega^{0:2m},\mathcal{D}')} \tag{16}$$

where $\omega^{0:t} := \{\omega^s\}_{s=1}^t$ and $\Omega^t$ denotes random variable of $\omega^t$. Since $\omega^0$ is randomly generated, which is independent of dataset. We have $P_{\Omega^0}(\omega^0|\mathcal{D}) = P_{\Omega^0}(\omega^0|\mathcal{D}')$.

Consider the $(2m+1)$-th iteration, by first-order condition, we have:

$$n^m = -\nabla F_i(\omega^{2m+1};\mathcal{D}) - \mu(\omega^{2m+1} - \overline{\omega}^{2m}) := g(\omega^{2m+1};\mathcal{D})$$

Given $\omega^{0:2m}$, $n^m$ and $\omega^{2m+1}$ will be bijective and the relation is captured by a one-to-one mapping $g : \mathbb{R}^d \to \mathbb{R}^d$ defined above. By Jacobian transformation, we have

$$P_{\Omega^{2m+1}}(\omega^{2m+1}|\omega^{0:2m},\mathcal{D}) = P_{N^m}(g(\omega^{2m+1};\mathcal{D})) \cdot |\det(\mathbf{J}(g(\omega^{2m+1};\mathcal{D})))|$$

Therefore,

$$\frac{P_{\Omega^{2m+1}}(\omega^{2m+1}|\omega^{0:2m},\mathcal{D})}{P_{\Omega^{2m+1}}(\omega^{2m+1}|\omega^{0:2m},\mathcal{D}')} = \frac{P_{N^m}(g(\omega^{2m+1};\mathcal{D}))}{P_{N^m}(g(\omega^{2m+1};\mathcal{D}'))} \cdot \frac{|\det(\mathbf{J}(g(\omega^{2m+1};\mathcal{D})))|}{|\det(\mathbf{J}(g(\omega^{2m+1};\mathcal{D}')))|}$$

Let $n^m := g(\omega^{2m+1};\mathcal{D})$, $n^{m'} := g(\omega^{2m+1};\mathcal{D}')$ be noise vectors that result in output $\omega^{2m+1}$ under neighboring datasets $\mathcal{D}$ and $\mathcal{D}'$ respectively. WLOG, let $d_1 \in \mathcal{D}$ and $d_1' \in \mathcal{D}'$ be the data pints in two datasets that are different, and $\mathcal{D} \setminus d_1 = \mathcal{D}' \setminus d_1'$. Because noise vector $N^m \sim \exp(-\alpha^m||n^m||)$, we have,

$$\frac{P_{N^m}(g(\omega^{2m+1};\mathcal{D}))}{P_{N^m}(g(\omega^{2m+1};\mathcal{D}'))} \le \exp(\alpha^m||n^m - n^{m'}||) = \exp(\alpha^m||\nabla F_i(\omega^{2m+1};\mathcal{D}') - \nabla F_i(\omega^{2m+1};\mathcal{D})||)$$

$$= \exp\left(\frac{\alpha^m}{|\mathcal{D}|}||\nabla F_i(\omega^{2m+1};d_1') - \nabla F_i(\omega^{2m+1};d_1)||\right) \le \exp\left(\frac{2\alpha^m u_1}{|\mathcal{D}|}\right) \tag{17}$$

Jacobian matrix

$$\mathbf{J}(g(\omega^{2m+1};\mathcal{D}))) = -\nabla^2 F_i(\omega^{2m+1};\mathcal{D}) - \mu\mathbf{I}_d := A \tag{18}$$

Further define matrix

$$A_\Delta = \mathbf{J}(g(\omega^{2m+1};\mathcal{D}'))) - A = \frac{1}{|\mathcal{D}|}\left(\nabla^2 F_i(\omega^{2m+1};d_1) - \nabla^2 F_i(\omega^{2m+1};d_1')\right)$$

Then

$$\frac{|\det(\mathbf{J}(g(\omega^{2m+1};\mathcal{D})))|}{|\det(\mathbf{J}(g(\omega^{2m+1};\mathcal{D}')))|} = \frac{|\det(A)|}{|\det(A_\Delta + A)|} = \frac{1}{|\det(I + A^{-1}A_\Delta)|} = \frac{1}{|\prod_{k=1}^r(1 + \lambda_k(A^{-1}A_\Delta))|}$$

where $\lambda_k(A^{-1}A_\Delta)$ denotes the $k$-th largest eigenvalue of matrix $A^{-1}A_\Delta$. Under generalized linear models, $A_\Delta$ has rank at most 2. Because $-\frac{u_2}{|\mathcal{D}|\mu} \leq \lambda_k(A^{-1}A_\Delta) \leq \frac{u_2}{|\mathcal{D}|\mu}$ and $\mu, u_2, |\mathcal{D}|$ satisfy $\frac{u_2}{|\mathcal{D}|\mu} \leq 0.5$, we have,

$$\frac{|\det(\mathbf{J}(g(\omega^{2m+1}; \mathcal{D})))|}{|\det(\mathbf{J}(g(\omega^{2m+1}; \mathcal{D}')))|} \leq \frac{1}{|1 - \frac{u_2}{|\mathcal{D}|\mu}|^2} = \exp(-2\ln(1 - \frac{u_2}{|\mathcal{D}|\mu})) \leq \exp\left(\frac{2.8u_2}{|\mathcal{D}|\mu}\right) \tag{19}$$

where the last inequality holds because $-\ln(1-x) < 1.4x$, $\forall x \in [0, 0.5]$.

Combine Eqn. (16), (19) and (17), we have

$$
\begin{aligned}
\frac{P_{\Omega^{0:2M}}(\omega^{0:2M}|\mathcal{D})}{P_{\Omega^{0:2M}}(\omega^{0:2M}|\mathcal{D}')} &\leq \prod_{m=0}^{M} \exp\left(\frac{2\alpha^m u_1}{|\mathcal{D}|}\right) \cdot \exp\left(\frac{2.8u_2}{|\mathcal{D}|\mu}\right) \\
&= \exp\left(\sum_{m=0}^{M} \frac{2\alpha^m u_1 \mu + 2.8u_2}{|\mathcal{D}|\mu}\right)
\end{aligned}
$$

Theorem 6.2 is proved. □

# F   Experiments

## F.1   Details of Datasets

Table 2: Details of datasets. Numbers in parentheses represent the amount of test data. All of the numbers round to integer.

|  | Dataset | Samples | # of device | Samples per device | |
|---|---|---|---|---|---|
|  |  |  |  | mean | stdev |
| Syn | iid | 6726(683) | 30 | 224 | 166 |
|  | 0,0 | 13791(1395) | 30 | 460 | 841 |
|  | 0.5,0.5 | 8036(818) | 30 | 268 | 410 |
|  | 1,1 | 10493(1063) | 30 | 350 | 586 |
| FEMNIST |  | 16421(1924) | 50 | 328 | 273 |
| Sent140 |  | 32299(8484) | 52 | 621 | 105 |

**Synthetic.** The synthetic data is generated using the same method in Li et al. (2020). We briefly describe the generating steps here. For each device k, $y_k$ is computed from a softmax function $y_k = \text{argmax}(\text{softmax}(W_k x_k + b_k))$. $W_k$ and $b_k$ are drawn from the same Gaussian distribution with mean $u_k$ and variance 1, where $u_k \in N(0; \beta)$. $x_k \in N(v_k; \Sigma)$. $v_k$ is drawn from a Gaussian distribution with mean $B_k \in \mathcal{N}(0, \gamma)$ and variance 1. $\Sigma$ is diagonal with $\sum j, j = j^{-1.2}$. In such a setting, $\beta$ controls how many local models differ from each other and $\gamma$ controls how much local data at each device differs from that of other devices.

In our experiment, we take $k = 30$, $x \in \mathcal{R}^{20}$, $W \in \mathcal{R}^{10*20}$, $b \in \mathcal{R}^{10}$. We generate 4 datasets in total. They're Syn(iid) Syn(0,0) with $\beta = 0$ and $\gamma = 0$, Syn(0.5,0.5) with $\beta = 0.5$ and $\gamma = 0.5$ and Syn(1,1) with $\beta = 1$ and $\gamma = 1$. In the output perturbation experiments, we set $\sigma$ to 1.0 for the baseline methods and to 0.8 for the Upcycled baselines, ensuring that the privacy budget $\epsilon$ for the baselines is always greater than that for the Upcycled baselines (e.g., the baseline $\bar{\epsilon} = 0.773$ and the Upcycled baseline $\bar{\epsilon} = 0.683$ for Syn(iid)). In the objective perturbation experiments, we set $\alpha$ to 10 for the baselines and 20 for the Upcycled baselines to achieve similar levels of information leakage, while still maintaining that $\epsilon$ for the Upcycled baselines is less than for the others. This constraint is maintained across all experiments.

**FEMNIST:** Similar with Li et al. (2020), we subsample 10 lower case characters ('a'-'j') from EMNIST Cohen et al. (2017) and distribute 5 classes to each device. There are 50 devices in total. The input is 28x28 image. In the privacy experiments, $\sigma$ is set to 0.27 for the baselines and 0.2 for the Upcycled baselines, and $\alpha$ is set to 100 for the baselines and 200 for the Upcycled baselines, respectively.

Sent140: A text sentiment analysis task on tweets Go et al. (2009). The input is a sequence of length 25 and the output is the probabilities of 2 classes. Here, $\sigma$ is set to 0.27 for the baselines and 0.2 for the Upcycled baselines, and $\alpha$ is set to 15 and 30 respectively.

A brief summary of dataset can be found in Table 2.

### F.2   Details of Algorithm `FedProx`

Here we present the detailed algorithm of `FedProx`:

---

**Algorithm 2** `FedProx` (Li et al., 2020)

---

1: **Input:** $\mu > 0$, $\{\mathcal{D}_i\}_{i \in \mathcal{I}}$, $\overline{\omega}^0$
2: **for** $t = 1$ **to** $T$ **do**
3:     The central server sends the current global model parameter $\overline{\omega}^t$ to all the clients.
4:     A subset of clients get active and each active client updates its local model by finding (approximate) minimizer of local loss function:

$$\omega_i^{t+1} = \arg\min_\omega F_i(\omega; \mathcal{D}_i) + \frac{\mu}{2}||\omega - \overline{\omega}^t||^2.$$

5:     Each client sends its local model to server.
6:     The central server updates the global model by aggregating all local models:

$$\overline{\omega}^{t+1} = \sum_{i \in \mathcal{I}} p_i \omega_i^{t+1}.$$

7: **end for**

---

### F.3 Convergence on all datasets

Here we present the convergence and accuracy of the testing dataset in the final iteration for all datasets under 90% straggler and 30% straggler scenarios.

### F.3.1 90% Straggler

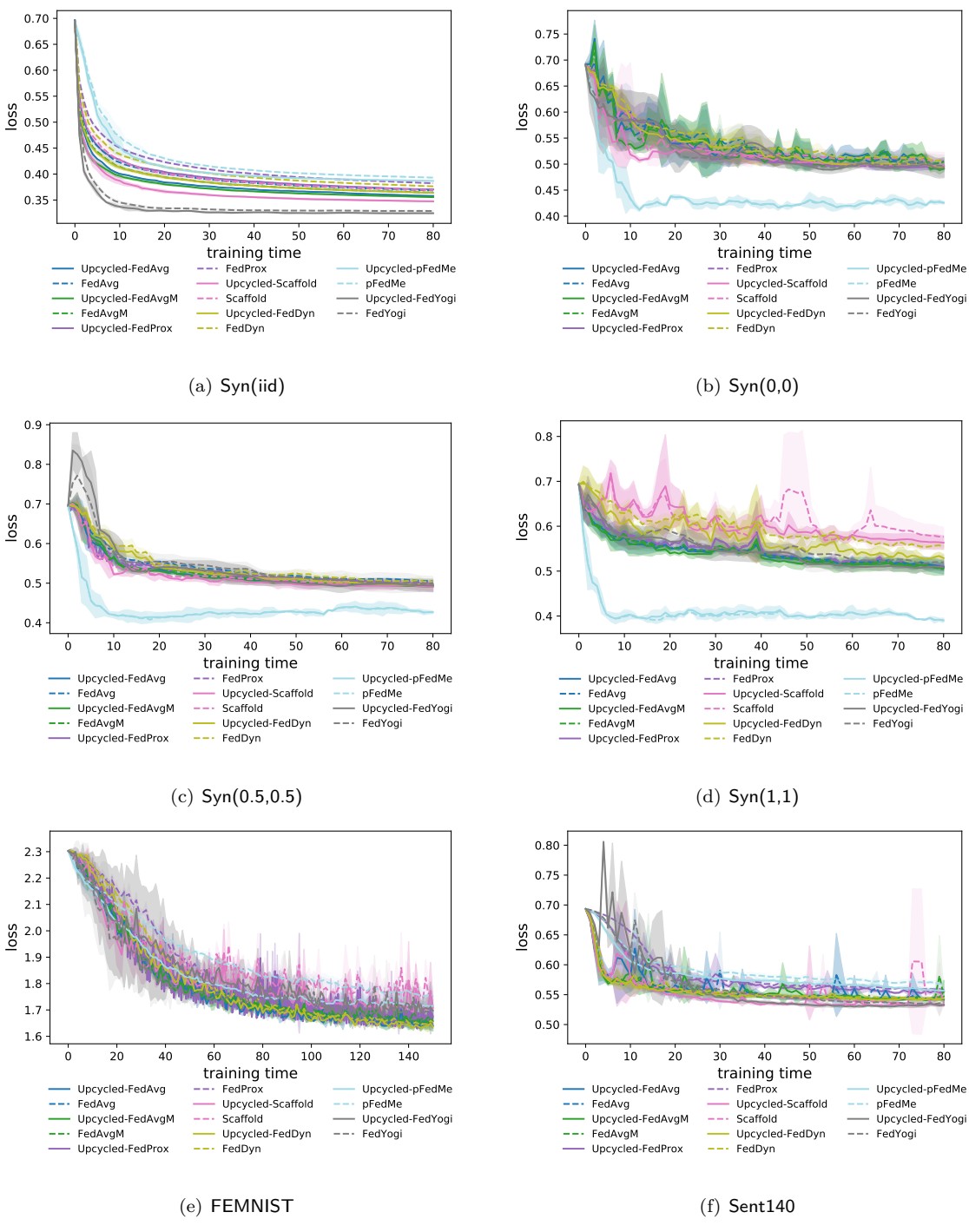

(a) Syn(iid)

(b) Syn(0,0)

(c) Syn(0.5,0.5)

(d) Syn(1,1)

(e) FEMNIST

(f) Sent140

Figure 5: Convergence of `Upcycled-FL` and regular FL methods with the approximate same training time under 90% Straggler.

### F.3.2    30% Straggler

We also conduct an experiment with a 30% straggler scenario to examine the convergence of our method. Table 3 demonstrates that our conclusions remain valid even when the straggler rate is relatively low.

Table 3: Average accuracy with 30% straggler on the testing dataset over four runs, the experimental setting is same as Table 1.

| Method | Dataset | | | | | |
|---|---|---|---|---|---|---|
| | Syn(iid) | Syn(0,0) | Syn(0.5,0.5) | Syn(1,1) | FEMNIST | Sent140 |
| FedAvg | $98.50_{\pm0.18}$ | $80.30_{\pm1.01}$ | $82.60_{\pm0.72}$ | $78.81_{\pm2.10}$ | $83.06_{\pm1.30}$ | $74.29_{\pm0.12}$ |
| **Upcycled-FedAvg** | $99.30_{\pm0.25}$ | $80.77_{\pm1.09}$ | $83.09_{\pm0.14}$ | $80.74_{\pm2.71}$ | $84.41_{\pm0.09}$ | $75.18_{\pm0.50}$ |
| FedAvgM | $98.50_{\pm0.07}$ | $81.08_{\pm2.33}$ | $82.89_{\pm0.12}$ | $80.64_{\pm2.64}$ | $81.60_{\pm4.57}$ | $74.61_{\pm2.28}$ |
| **Upcycled-FedAvgM** | $99.12_{\pm0.17}$ | $82.20_{\pm1.80}$ | $82.68_{\pm0.19}$ | $80.43_{\pm2.51}$ | $82.31_{\pm3.17}$ | $74.16_{\pm2.84}$ |
| FedProx | $97.22_{\pm0.24}$ | $80.88_{\pm0.67}$ | $82.31_{\pm1.05}$ | $80.13_{\pm2.76}$ | $81.57_{\pm1.46}$ | $74.23_{\pm0.31}$ |
| **Upcycled-FedProx** | $98.24_{\pm0.21}$ | $81.52_{\pm0.82}$ | $83.05_{\pm0.14}$ | $81.73_{\pm0.84}$ | $82.85_{\pm1.60}$ | $74.83_{\pm0.14}$ |
| Scaffold | $98.28_{\pm0.14}$ | $79.95_{\pm1.67}$ | $79.54_{\pm0.72}$ | $72.27_{\pm4.68}$ | $76.85_{\pm5.55}$ | $75.76_{\pm0.21}$ |
| **Upcycled-Scaffold** | $99.52_{\pm0.18}$ | $79.46_{\pm1.62}$ | $81.83_{\pm2.08}$ | $73.34_{\pm1.13}$ | $77.69_{\pm2.39}$ | $77.01_{\pm0.30}$ |
| FedDyn | $98.17_{\pm0.08}$ | $82.06_{\pm0.61}$ | $80.97_{\pm1.04}$ | $78.65_{\pm3.83}$ | $84.39_{\pm0.95}$ | $75.88_{\pm0.11}$ |
| **Upcycled-FedDyn** | $98.57_{\pm0.22}$ | $82.08_{\pm0.68}$ | $82.80_{\pm1.42}$ | $81.11_{\pm2.64}$ | $85.34_{\pm2.47}$ | $75.90_{\pm0.33}$ |
| pFedme | $96.45_{\pm0.14}$ | $91.05_{\pm0.60}$ | $89.64_{\pm1.04}$ | $92.88_{\pm0.67}$ | $76.15_{\pm3.08}$ | $72.87_{\pm0.26}$ |
| **Upcycled-pFedme** | $96.89_{\pm0.27}$ | $91.06_{\pm0.61}$ | $89.40_{\pm1.13}$ | $92.64_{\pm0.90}$ | $77.36_{\pm3.59}$ | $73.48_{\pm0.49}$ |
| FedYogi | $99.38_{\pm0.28}$ | $81.06_{\pm2.53}$ | $80.65_{\pm1.32}$ | $79.14_{\pm1.63}$ | $79.98_{\pm7.83}$ | $77.63_{\pm0.29}$ |
| **Upcycled-FedYogi** | $99.60_{\pm0.14}$ | $81.77_{\pm2.06}$ | $81.72_{\pm1.02}$ | $80.32_{\pm0.76}$ | $81.96_{\pm4.28}$ | $77.78_{\pm0.85}$ |

### F.3.3 30% Straggler

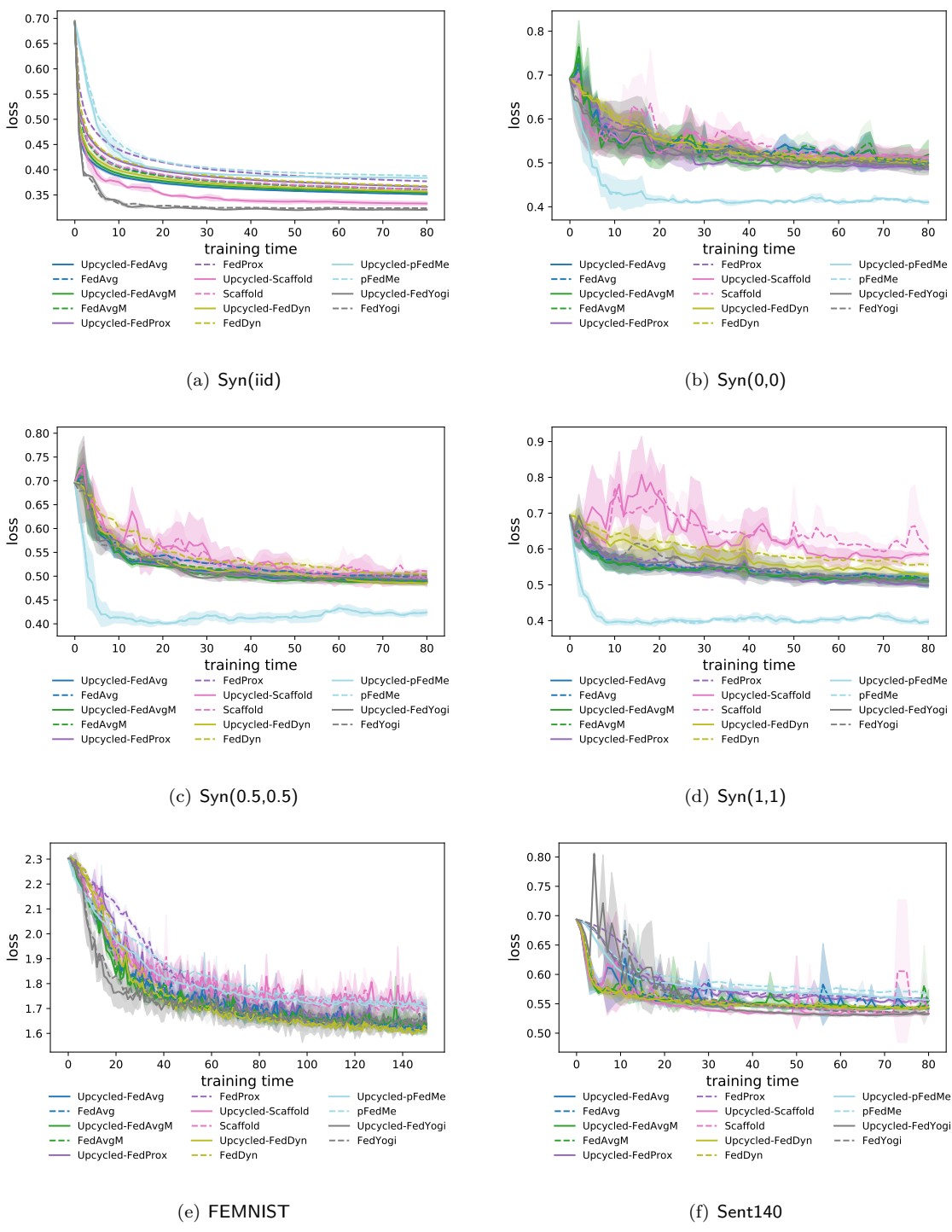

Figure 6: Convergence of `Upcycled-FL` and regular methods with the approximate same training time under 30% Straggler.

## F.4    Additional Privacy Experiments

In addition to privacy experiments on synthetic datasets, we also report the output perturbation and objective perturbation on real-world datasets: FEMNIST and Sent140 in Figure 7 and 8. Due to limited computational resources, we conduct these experiments using a subset of the previously established baselines.

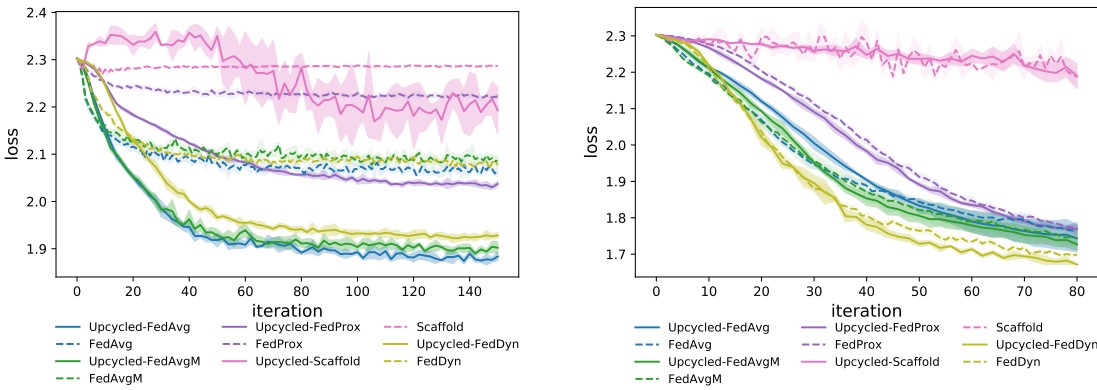

Figure 7: Comparison of private `Upcycled-FL` and private FL methods using **output perturbation** (left) and **objective perturbation** (right) on FEMNIST.

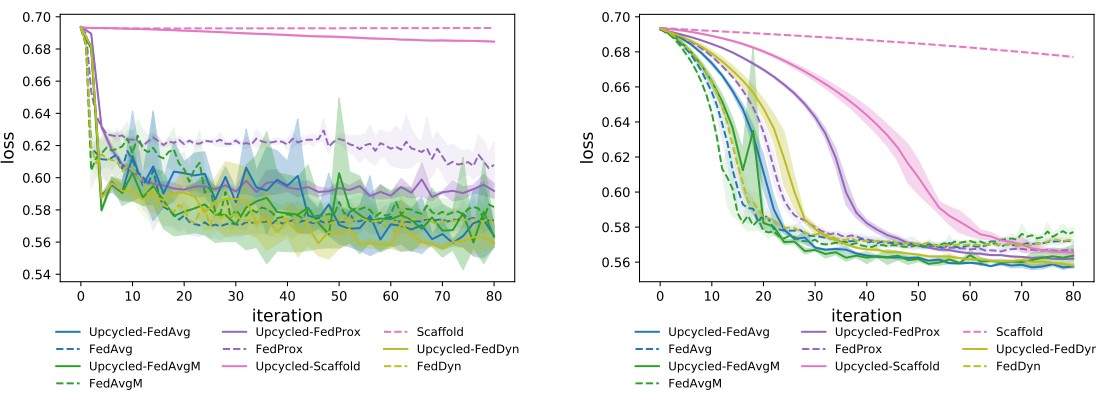

Figure 8: Comparison of private `Upcycled-FL` and private FL methods using **output perturbation** (left) and **objective perturbation** (right) on Sent140.

## F.5    Experiments on full-scale lowercase letter of FEMNIST

We conduct a convergence experiment with full-scale lowercase letters (26 classes) from the FEMNIST dataset, as shown in Table 4. We also conduct privacy experiments comparing our method with FedProx, as shown in Figure 9.

Table 4: Average accuracy on the large-scale FEMNIST dataset

| Method | FEMNIST(lowercase letters) | |
|---|---|---|
| | 30% straggler | 90% straggler |
| FedAvg | $90.90 \pm 4.55$ | $88.50 \pm 5.07$ |
| **Upcycled-FedAvg** | $91.26 \pm 4.06$ | $88.69 \pm 4.94$ |
| FedAvgM | $91.02 \pm 4.47$ | $89.13 \pm 4.35$ |
| **Upcycled-FedAvgM** | $91.05 \pm 4.36$ | $89.28 \pm 4.05$ |
| FedProx | $91.91 \pm 0.38$ | $89.02 \pm 4.53$ |
| **Upcycled-FedProx** | $92.08 \pm 0.72$ | $89.44 \pm 4.30$ |
| Scaffold | $89.69 \pm 3.92$ | $89.11 \pm 3.74$ |
| **Upcycled-Scaffold** | $89.77 \pm 4.01$ | $89.72 \pm 3.90$ |
| FedDyn | $92.88 \pm 0.30$ | $91.42 \pm 0.24$ |
| **Upcycled-FedDyn** | $93.56 \pm 0.30$ | $92.64 \pm 0.37$ |
| pFedme | $81.36 \pm 4.63$ | $80.72 \pm 1.47$ |
| **Upcycled-pFedme** | $85.62 \pm 1.43$ | $85.14 \pm 0.93$ |
| FedYogi | $87.64 \pm 1.05$ | $85.61 \pm 0.56$ |
| **Upcycled-FedYogi** | $88.89 \pm 1.84$ | $86.48 \pm 1.30$ |

Table 5: Training time for output perturbation experiments on Syn(iid).

| Time/s | FedAvg | FedAvgM | FedProx | Scaffold | FedDyn | pFedme | FedYogi |
|---|---|---|---|---|---|---|---|
| Baseline | $147.50 \pm 6.69$ | $147.47 \pm 7.49$ | $139.91 \pm 10.81$ | $153.88 \pm 12.70$ | $181.49 \pm 17.60$ | $467.42 \pm 12.87$ | $153.53 \pm 13.72$ |
| Upcycled | $86.44 \pm 9.23$ | $84.95 \pm 3.39$ | $80.59 \pm 3.29$ | $95.09 \pm 13.89$ | $99.82 \pm 3.32$ | $259.54 \pm 18.27$ | $92.82 \pm 2.24$ |

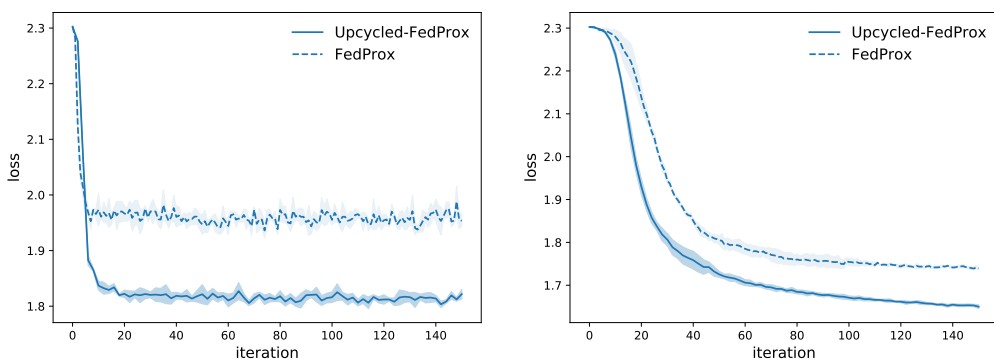

Figure 9: Comparison of private `Upcycled-FedProx` and private `FedProx` using **output perturbation** (left) and **objective perturbation** (right).

## F.6 Training Time

We report the average training time to compare the communication cost in Table 5.

