# OpenReview forum: "Federated Learning with Reduced Information Leakage and Computation"
_TMLR — Accepted by TMLR_

### Review · Reviewer_4RHt · 2024-04-07

**Summary Of Contributions:**

This paper proposes a Federated Learning (FL) strategy to minimize information leakage. It achieves this by accessing data only in odd rounds, while for even rounds, it updates the model by applying a first-order approximation based on the previous rounds’ model.
The proposed technique looks simple but effective.

**Audience:**

Yes

**Broader Impact Concerns:**

There is no concerns on the ethical implications of the work.

**Claims And Evidence:**

Yes

**Requested Changes:**

There are a few comments.

1. Predicting the next update by ‘upcycling information’ may be less accurate as the data held by each client is non-iid. However, the experimental results seem to show that, on the contrary, there is further improvement. Could you explain me the reason why that is?

2. 30% of devices are selected in each iteration. Does the proposed technique still show better results even when fewer devices are selected?

3. Why upcycle only in even iteration? Isn't it possible to do it at different intervals, like 3 times or 4 times? How would accuracy or convergence change in that case?

4. Figures are too difficult to recognize. Please think about ways to presenst the experiment results more clearly.

5. "More rlated work is discussed in Appendix 2" in page 2, but thre is no Appendix 2.

**Strengths And Weaknesses:**

Strength: The improvement is achieved with a simple technique and can be applied to general FL algorithms.

Weakness: There is a lack of experiments on various dataset and the improvement achieved by the proposed technique is not significant.

---

> ### Author Response · Authors · 2024-05-04
>
> # To Reviewer 4RHt
>
> We thank the reviewer for the constructive suggestions. We address each of your concerns in the subsequent response.
>
> ### RE: Weakness: There is a lack of experiments on various dataset and the improvement achieved by the proposed technique is not significant.
>
> We have included another baseline and a new experiment on lowercase letters of the FEMNIST dataset, which consists of more than 100,000 samples across 26 classes. As expected, our method still consistently outperforms the baselines.
>
> Regarding the improvement attained under the proposed technique, note that Upcycled-FL is compared with baselines under two settings: 1) In DP settings when the perturbation is needed, our method can **significantly outperform** baselines regarding privacy-accuracy tradeoff (especially when privacy requirement is high). As shown in Figure 3(a), the Upcycled strategy significantly reduces the loss value for all baseline methods within half the training time and with a smaller privacy budget; 2) Even for non-private settings, ours can outperform baselines under similar training time. Although the improvement is not as significant as in private settings, we emphasize that our proposed strategy is a very simple plug-in method that can be directly adapted to any FL algorithm.
>
> ### RE: Predicting the next update by ‘upcycling information’ may be less accurate as the data held by each client is non-iid. However, the experimental results seem to show that, on the contrary, there is further improvement. Could you explain me the reason why that is?
>
> In our method, even updates $\omega_i^{2m}$ are derived directly from the gradients $\nabla F_i(\omega_i^{2m-1};\mathcal{D}_i)$ in previous odd iteration. It means data samples used for odd updates is upcycled and reused for even updates, i.e., odd and even updates are computed from the same batch of clients.
>
> The reasons why our method can outperform the baselines are as follows: 1) compared to baselines, the same data samples selected for odd updates are upcycled and used again for even updates. Because even updates only involve addition/subtraction operations, under similar training time, our method outperforms baselines by updating the global model more times (Fig. 3); 2) When noises are needed to ensure DP, our method only needs less perturbation to ensure the same privacy guarantee (because information leakage only incurs during half of iterations). With less added noise, ours can attain much higher accuracy compared to the baselines (Fig. 5 \& 6).
>
> ### 30\% of devices are selected in each iteration. Does the proposed technique still show better results even when fewer devices are selected?
>
> Yes, our method consistently outperforms baselines under varying fractions of active clients. As an example, we report the results below when 10\% devices are selected in each iteration.
>
> | Method            | Syn(iid)           | Syn(0,0)           | Syn(0.5,0.5)       | Syn(1,1)           | FEMNIST            | Sent140            |
> | ----------------- | ------------------ | ------------------ | ------------------ | ------------------ | ------------------ | ------------------ |
> | FedAvg            | $97.36 \pm 0.38$ | $78.76 \pm 0.67$ | $80.50 \pm 1.52$ | $77.23 \pm 2.35$ | $79.60 \pm 3.32$ | $75.32 \pm 1.18$ |
> | Upcycled-FedAvg   | $98.32 \pm 0.19$ | $80.77 \pm 1.20$ | $82.70 \pm 1.03$ | $78.34 \pm 1.78$ | $81.77 \pm 2.88$ | $76.05 \pm 0.38$ |
> | FedAvgM           | $98.10 \pm 0.27$ | $79.71 \pm 2.73$ | $82.00 \pm 0.73$ | $77.70 \pm 3.51$ | $79.00 \pm 3.15$ | $76.01 \pm 0.40$ |
> | Upcycled-FedAvgM  | $98.54 \pm 0.41$ | $81.18 \pm 1.65$ | $82.76 \pm 1.08$ | $78.36 \pm 1.27$ | $80.56 \pm 3.59$ | $76.38 \pm 0.38$ |
> | FedProx           | $96.12 \pm 0.30$ | $78.94 \pm 0.66$ | $81.45 \pm 1.15$ | $77.61 \pm 3.35$ | $80.86 \pm 0.77$ | $73.79 \pm 0.37$ |
> | Upcycled-FedProx  | $97.29 \pm 0.39$ | $79.62 \pm 0.82$ | $82.76 \pm 1.01$ | $78.72 \pm 2.51$ | $82.09 \pm 1.32$ | $74.37 \pm 0.88$ |
> | Scaffold          | $97.62 \pm 0.14$ | $76.13 \pm 1.10$ | $77.91 \pm 3.78$ | $67.04 \pm 2.38$ | $67.04 \pm 2.38$ | $74.45 \pm 0.27$ |
> | Upcycled-Scaffold | $97.80 \pm 0.17$ | $76.02 \pm 0.99$ | $77.14 \pm 3.19$ | $70.74 \pm 1.49$ | $70.74 \pm 1.49$ | $76.41 \pm 0.50$ |
> | FedDyn            | $96.67 \pm 0.39$ | $79.14 \pm 0.51$ | $81.14 \pm 0.90$ | $76.43 \pm 4.52$ | $82.41 \pm 0.86$ | $75.85 \pm 0.56$ |
> | Upcycled-FedDyn   | $98.10 \pm 0.21$ | $79.82 \pm 0.76$ | $81.78 \pm 1.21$ | $76.93 \pm 5.59$ | $83.46 \pm 0.62$ | $75.95 \pm 0.17$ |
> | pFedme            | $94.95 \pm 0.45$ | $87.88 \pm 3.62$ | $88.61 \pm 2.39$ | $91.66 \pm 1.68$ | $75.98 \pm 4.55$ | $73.57 \pm 0.61$ |
> | Upcycled-pFedme   | $96.26 \pm 0.39$ | $87.94 \pm 3.27$ | $88.54 \pm 1.85$ | $91.55 \pm 1.89$ | $81.09 \pm 2.01$ | $73.73 \pm 0.26$ |

---

> > ### Author Response · Authors · 2024-05-04
> >
> > ### RE: Why upcycle only in even iteration? Isn't it possible to do it at different intervals, like 3 times or 4 times? How would accuracy or convergence change in that case?
> >
> > In our method, client's local gradient information is upcycled via *first-order approximation*. By replacing original objective function $F_i(\omega;\mathcal{D}_i)$ with Taylor series expansion Eqn. (2), even updates $\omega_i^{2m}$ can be derived directly from gradient $\nabla F_i(\omega_i^{2m-1};\mathcal{D}_i)$ in previous odd iteration. One reason our method does even/odd alternation is that by minimizing the actual local objective function in odd iterations, the gradient $\nabla F_i(\omega_i^{2m-1};\mathcal{D}_i)$ can be attained efficiently with only addition and subtraction operation using *first-order condition* (e.g., $\nabla F_i(\omega_i^{2m-1};\mathcal{D}_i) = -\mu(\omega_i^{2m-1}-\overline{\omega}^{2m-2})$ as shown Eqn. (4)), without computing derivatives.
> >
> > If we change the frequency to 3, i.e., apply first-order approximation two times consecutively in $(3m+1)$-th and $(3m+2)$-th iterations after minimizing local objective in $3m$-th iteration, then the gradient $\nabla F_i(\omega_i^{3m+1};\mathcal{D}_i)$ used for updating $\omega_i^{3m+2}$ can no longer be computed efficiently using addition/subtraction operations. This is because $\omega_i^{3m+1}$ is not the minimum of local objective function and does not satisfy first-order condition anymore.
> >
> > ### RE: Figures are too difficult to recognize. Please think about ways to presenst the experiment results more clearly.
> >
> > We have updated the figures to display the results more clearly.
> >
> > ### RE: "More related work is discussed in Appendix 2" in page 2, but there is no Appendix 2.
> >
> > We thank the reviewer for pointing it out. It is a typo; more related work is discussed in Section 2.

---

### Review · Reviewer_QNCJ · 2024-04-11

**Summary Of Contributions:**

This paper proposes Upcycled-FL, a simple yet effective method to save the communication cost and information leakage. The main idea is to reduce the communication rounds by updating the even rounds with previous updates from first-order expansion. The main method is provided with convergence analysis, privacy guarantees and comparison with multiple baselines.

**Audience:**

Yes

**Broader Impact Concerns:**

I don't have broader impact concerns.

**Claims And Evidence:**

Yes

**Requested Changes:**

See above.

**Strengths And Weaknesses:**

Strengths:
1. the paper proposes a simple method that seems to work well, with clear motivation and explanation.
2. The algorithm is supported with convergence guarantees and privacy analysis.
3. The method is compared with a lot of baseline FL algorithms.

Weaknesses:
1. Only two examples are provided in Section 4 (FedAvg and FedProx). The authors should add a general modification description of FL methods.
2. The method could be more general. Instead of only even/odd alternation, can we think about more general update rules? For example, we can change the frequency to 3 or more (i.e., communicate every 3 updates).
3. Line 6 of Algorithm 1 seems like a momentum mechanism or extrapolation. The authors may want to explain the connection to existing optimization theory on this.
4. Assumptions 5.2, 5.4 are standard but may not be practical. Could the authors explain how they are satisfied in your experiments?
5. In addition to the convergence and privacy analysis, I think communication cost is another important factor. There should be some analysis on the communication.
6. The main privacy results are in Appendix B, which I think are important and should be discussed in the main text.
7. The main experiment datasets are either synthetic or small. The authors should try larger scale datasets for federated learning.
8. The improvement after adding Upcycle seems a bit minor. This shows that Upcycle-FL may not be very effective.
9. Missing baseline for FedYogi (https://arxiv.org/abs/2003.00295).
10. Missing a table comparing the communication cost although partially reflected in Table 1.
11. Privacy experiments are only done on synthetic datasets. What about standard datasets like FEMNIST and Sent140?

---

> ### Author Response · Authors · 2024-05-04
>
> # To Reviewer QNCJ
>
> We thank the reviewer for the valuable comments.  Here we address your concerns:
>
> ### Only two examples are provided in Section 4 (FedAvg and FedProx). The authors should add a general modification description of FL methods.
>
> We want to emphasize our proposed method is general; the idea is to upcycle client's local gradient information via *first-order approximation*, i.e., replacing original objective function $F_i(\omega;\mathcal{D}_i)$ with Taylor series expansion Eqn.~(2), and compute even updates $\omega_i^{2m}$ directly from gradient $\nabla F_i(\omega_i^{2m-1};\mathcal{D}_i)$ in previous odd iteration (without accessing local dataset $\mathcal{D}_i$ in even iterations). This method is a plug-in and can be applied to many existing FL algorithms; its general description was given in Section 4 (lines 10 - 19). We use two classic FL algorithms (FedAvg and FedProx) as examples to demonstrate the proposed Upcycled-FL.
>
> ### RE: The method could be more general. Instead of only even/odd alternation, can we think about more general update rules? For example, we can change the frequency to 3 or more (i.e., communicate every 3 updates).
>
> In our method, client's local gradient information is upcycled via *first-order approximation*. By replacing original objective function $F_i(\omega;\mathcal{D}_i)$ with Taylor series expansion Eqn. (2), even updates $\omega_i^{2m}$ can be derived directly from gradient $\nabla F_i(\omega_i^{2m-1};\mathcal{D}_i)$ in previous odd iteration. One reason our method does even/odd alternation is that by minimizing the actual local objective function in odd iterations, the gradient $\nabla F_i(\omega_i^{2m-1};\mathcal{D}_i)$ can be attained efficiently with only addition and subtraction operation using *first-order condition* (e.g., $\nabla F_i(\omega_i^{2m-1};\mathcal{D}_i) = -\mu(\omega_i^{2m-1}-\overline{\omega}^{2m-2})$ as shown Eqn. (4)), without computing derivatives.
>
> If we change the frequency to 3, i.e., apply first-order approximation two times consecutively in $(3m+1)$-th and $(3m+2)$-th iterations after minimizing local objective in $3m$-th iteration, then the gradient $\nabla F_i(\omega_i^{3m+1};\mathcal{D}_i)$ used for updating $\omega_i^{3m+2}$ can no longer be computed efficiently using addition/subtraction operations. This is because $\omega_i^{3m+1}$ is not the minimum of local objective function and does not satisfy first-order condition anymore.
>
> ### RE: Line 6 of Algorithm 1 seems like a momentum mechanism or extrapolation. The authors may want to explain the connection to existing optimization theory on this.
>
> Momentum mechanism considers the past gradients to smooth the updates; it uses historical information from multiple previous iterations. In contrast, Upcycled-FL only leverages the update in the preceding iteration to optimize the model in the even iteration. Indeed, our method of "upcycling gradient information via first-order approximation" can result in the global model updates at even iterations being reduced to the form of linear extrapolation.
>
> ### RE: Assumptions 5.2, 5.4 are standard but may not be practical. Could the authors explain how they are satisfied in your experiments?
>
> Both assumptions can be satisfied under certain loss functions and hypothesis classes, e.g., experiments of logistic regression on synthetic datasets. However, we emphasize that Assumptions 5.2 and 5.4 are primarily used in theoretical analysis and they serve as **sufficient** (but **not necessary**) conditions for convergence guarantee. For the practical deployment of the algorithm, these conditions are not necessarily to hold. As shown in our experiments on real data, our method is still effective and outperforms the baselines even when these assumptions do not hold.
>
> ### RE: In addition to the convergence and privacy analysis, I think communication cost is another important factor. There should be some analysis on the communication.
>
> With the same training epochs, Upcycled-FL requires half of the communication cost compared to traditional FL methods (as we don't send and receive data in even iterations). We have updated the manuscript and discussed the communication cost in the discussion part of Section 4 and the Table 5 in Appendix.
>
> ### RE: The main privacy results are in Appendix B, which I think are important and should be discussed in the main text.
>
> We have updated the manuscript and moved Appendix B to Section 6.

---

> > ### Author Response · Authors · 2024-05-04
> >
> > ### RE: The main experiment datasets are either synthetic or small. The authors should try larger scale datasets for federated learning.
> >
> > We added a new set of experiments on lowercase letters of the FEMNIST dataset, which consists of more than 100,000 samples across 26 classes. The results are added to the manuscript, as shown in Table 4 and Figure 9.
> >
> > Here we present the convergence results:
> >
> > | Method            | FEMNIST(100,000) 30% straggler | FEMNIST(100,000) 90% straggler |
> > | ----------------- | ------------------------------ | ------------------------------ |
> > | FedAvg            | $90.90 \pm 4.55$             | $88.50 \pm 5.07$             |
> > | Upcycled-FedAvg   | $91.26 \pm 4.06$             | $88.69 \pm 4.94$             |
> > | FedAvgM           | $91.02 \pm 4.47$             | $89.13 \pm 4.35$             |
> > | Upcycled-FedAvgM  | $91.05 \pm 4.36$             | $89.28 \pm 4.05$             |
> > | FedProx           | $91.91 \pm 0.38$             | $89.02 \pm 4.53$             |
> > | Upcycled-FedProx  | $92.08 \pm 0.72$             | $89.44 \pm 4.30$             |
> > | Scaffold          | $89.69 \pm 3.92$             | $89.11 \pm 3.74$             |
> > | Upcycled-Scaffold | $89.77 \pm 4.01$             | $89.72 \pm 3.90$             |
> > | FedDyn            | $92.88 \pm 0.30$             | $91.42 \pm 0.24$             |
> > | Upcycled-FedDyn   | $93.56 \pm 0.30$             | $92.64 \pm 0.37$             |
> > | pFedme            | $81.36 \pm 4.63$             | $80.72 \pm 1.47$             |
> > | Upcycled-pFedme   | $85.62 \pm 1.43$             | $85.14 \pm 0.93$             |
> > | FedYogi           | $87.64 \pm 1.05$             | $85.61 \pm 0.56$             |
> > | Upcycled-FedYogi  | $88.89 \pm 1.84$             | $86.48 \pm 1.30$             |
> >
> > ### RE: The improvement after adding Upcycle seems a bit minor. This shows that Upcycle-FL may not be very effective.
> >
> > Upcycled-FL is compared with baselines under two settings, with and without noises added for privacy protection. 1) In DP settings when the perturbation is needed, our method can **significantly outperform** baselines regarding privacy-accuracy tradeoff (especially when privacy requirement is high). For example, as shown in Figure 3(a), the Upcycled strategy significantly reduces the loss value for all baseline methods within HALF the training time and with a smaller privacy budget. **For example, when using output pertubation on FEMNIST, compared to FedDyn, Up-FedDyn achieves 8.45\% higher accuracy, 4.5\% smaller $\varepsilon$ (privacy budget), 49\% less training time;** 2) Even for non-private settings, ours can outperform baselines under similar training time. Although the improvement is not as significant as in private settings, we emphasize that our proposed strategy is a very simple plug-in method that can be directly adapted to any FL algorithm.
> >
> > ### RE: Missing baseline for FedYogi
> >
> > We have conducted another set of experiments on FedYogi and included the results in the updated Tables 1 and 3, and Figures 2-6.
> >
> > ### RE: Missing a table comparing the communication cost although partially reflected in Table 1.
> >
> > With the same training epochs, Upcycled-FL requires half of the communication cost compared to traditional FL methods (as we don't send and receive data in even iterations). We have updated the manuscript and discussed the commutation cost in the discussion part of Section 4 and the Table 5 in Appendix.
> >
> > ### RE: Privacy experiments are only done on synthetic datasets. What about standard datasets like FEMNIST and Sent140?
> >
> > Due to the page limit, the additional privacy experiments are provided in Figures 7 and 8 in the Appendix.

---

### Review · Reviewer_2WzD · 2024-04-22

**Summary Of Contributions:**

The paper proposes a new federated training method to reduce the communication cost and information leakage. Specifically, the paper proposes to only transfer the updates and merge on the odd iterations and uses the local model to do the update on the even iterations. The paper uses FedProx as an example to derive the upcycled version. The theoretical analysis are conduct to show the convergence rate of the proposed method and also briefly cover the privacy guarantee if differential private updates are utilized. The experimental results show the proposed upcycled FL could achieve a better convergence in both synthetic data and real-world data.

**Audience:**

Yes

**Broader Impact Concerns:**

None.

**Claims And Evidence:**

No

**Requested Changes:**

1. Have a more thorough and formal discussion about the information leakage problem.
2. Explain why the half information is the sweet point for the update but not with fewer synchronization iteration.
3. Experiments on real-world data with dirichlet distribution.

**Strengths And Weaknesses:**

Pros:
1. The paper is easy to follow and the proposed method is simple.
2. The experimental results show the proposed method is effective.

Cons:
1. The proposed updates is unclear to me in the cases that if the client is selected in even round but won't be selected in odd round. Will the model be updated if the client is not selected?
2.  I am not an expert on the differential privacy field. However, I believe the information leakage won't be reduced by simply half if only half of the updates need communication. If the assumption is true, the best strategy would be use at least as possible synchronization and the proposed method could be quickly adapted into using only 1/3, 1/4, ....
3.  The experiments results in Figure 2,3,4 seems weird. I could get the proposed method would be better in terms of training-time however  it could achieve a better rate in terms of iterations. The proposed method uses a less information but achieve a better convergence rate which does not make sense to me.
4. The experiments on the heterogeneous data distribution is not enough. The paper only uses synthetic data to simulate the heterogenous data distribution case. However, it would be more convincing if it could be done in the real-world data and it is also adopted by a lot of other federated learning methods to show their effectiveness. Partitioning the data with the dirichlet distribution would be a good way and the hyperparameter in the dirichlet distribution could be utilized to control the level of heterogeneity.

---

> ### Author Response · Authors · 2024-05-04
>
> # To Reviewer 2WzD
>
> We thank the reviewer for the evaluation of our work. We address the reviewer's concerns below:
>
> ### RE: The proposed updates is unclear to me in the cases that if the client is selected in even round but won't be selected in odd round. Will the model be updated if the client is not selected?
>
> In the proposed method, during each odd-numbered round, a subset of clients is selected to perform gradient updates. Then, in the subsequent even round, the server upcycles gradient information from the same subset of clients and updates the aggregated model without acquiring any additional information from clients.
>
> ### RE: I am not an expert on the differential privacy field. However, I believe the information leakage won't be reduced by simply half if only half of the updates need communication. If the assumption is true, the best strategy would be use at least as possible synchronization and the proposed method could be quickly adapted into using only 1/3, 1/4, ....
>
> In our method, client's local gradient information is upcycled via *first-order approximation*. By replacing original objective function $F_i(\omega;\mathcal{D}_i)$ with Taylor series expansion Eqn. (2), even updates $\omega_i^{2m}$ can be derived directly from gradient $\nabla F_i(\omega_i^{2m-1};\mathcal{D}_i)$ in previous odd iteration. One reason our method does even/odd alternation is that by minimizing the actual local objective function in odd iterations, the gradient $\nabla F_i(\omega_i^{2m-1};\mathcal{D}_i)$ can be attained efficiently with only addition and subtraction operation using *first-order condition* (e.g., $\nabla F_i(\omega_i^{2m-1};\mathcal{D}_i) = -\mu(\omega_i^{2m-1}-\overline{\omega}^{2m-2})$ as shown Eqn. (4)), without computing derivatives.
>
> If we change the frequency to 3, i.e., apply first-order approximation two times consecutively in $(3m+1)$-th and $(3m+2)$-th iterations after minimizing local objective in $3m$-th iteration, then the gradient $\nabla F_i(\omega_i^{3m+1};\mathcal{D}_i)$ used for updating $\omega_i^{3m+2}$ can no longer be computed efficiently using addition/subtraction operations. This is because $\omega_i^{3m+1}$ is not the minimum of local objective function and does not satisfy first-order condition anymore.
>
> ### RE: The experiments results in Figure 2,3,4 seems weird. I could get the proposed method would be better in terms of training-time however it could achieve a better rate in terms of iterations. The proposed method uses a less information but achieve a better convergence rate which does not make sense to me.
>
> Figures 2, 3, 4 compare our method with baselines in two settings.
>
> (1) Without differential privacy (DP) guarantee (Fig. 2): methods are compared under approximately
> the **same training time**. Here, the odd and even rounds of our method are *combined* as 1 round when plotting the figure. Since even updates only involve addition/subtraction operations, it takes approximately the same training time compared to 1 round of baselines. Because our method repeatedly uses the client's gradient information compared to baselines, ours has a better performance.
>
> (2) With DP (Fig. 3 \& 4): In this setting, our method uses approximately **half the training time** and still outperforms the baselines regarding both accuracy and privacy. This is because information leakage only occurs during odd iterations; because data $\mathcal{D}_i$ is only used half the rounds and less information is leaked, we can add less noise while still obtaining a smaller $\varepsilon$ (**stronger privacy**) than the baselines, and the less perturbation leads to **better accuracy**. In conclusion, when enabling DP, our method achieves better accuracy and privacy simultaneously while requiring significantly less training time.

---

> > ### Author Response · Authors · 2024-05-04
> >
> > ### RE: Have a more thorough and formal discussion about the information leakage problem.
> >
> > We have added the following discussion about the information leakage to the section 1 of the manuscript:
> >
> > When machine learning algorithms are trained on individual data, there is a risk of information leakage that a third party may infer individual sensitive information from (intermediate) computational outcomes.  In federated learning, information leakage occurs whenever the client's local gradients are shared with third parties. Importantly, the information leakage (or privacy loss) **accumulates** as data is repeatedly used during the iterative learning process: with more computational outcomes derived from individual data, third parties have more information to infer sensitive data and it poses higher privacy risks for individuals. An example is [1], which shows that eavesdroppers can conduct gradient inversion attacks to recover clients' data from the gradients. In this paper, we use the differential privacy framework to mathematically upper bound the information leakage, and each client's total privacy loss is quantified using the *composition theorem* of differential privacy. Because even updates $\omega_i^{2m}$ of our method can be derived directly from the gradients $\nabla F_i (\omega_i^{2m-1};\mathcal{D}_i)$ in previous odd iteration **without accessing data** $\mathcal{D}_i$, information leakage only occurs during odd iterations.
> >
> > [1] Huang, Yangsibo, et al. "Evaluating gradient inversion attacks and defenses in federated learning." Advances in Neural Information Processing Systems 34 (2021): 7232-7241.
> >
> > ### RE: Explain why the half information is the sweet point for the update but not with fewer synchronization iteration.
> >
> > The total privacy loss is quantified using the *composition theorem* outlined in Theorem 3.14 of "The Algorithmic Foundations of Differential Privacy." Our approach halves the data leakage because each client's data is only accessed in half of the rounds. Specifically, in our method, client's local gradient information is upcycled via *first-order approximation*. By replacing original objective function $F_i(\omega;\mathcal{D}_i)$ with Taylor series expansion Eqn. (2), even updates $\omega_i^{2m}$ can be derived directly from gradient $\nabla F_i(\omega_i^{2m-1};\mathcal{D}_i)$ in previous odd iteration without accessing data $\mathcal{D}_i$.
> >
> > We emphasize that our idea of "upcycling information" is fundamentally different from reducing the number of synchronizations.  While minimal synchronization generally results in less privacy leakage, it inevitably **compromises utility** because the global model is updated fewer times and less information is learned (i.e., there is an inherent privacy-accuracy trade-off). However, our method **improves this trade-off**: under the same privacy leakage, ours by upcycling data information can allow double rounds of model synchronization and result in a higher model accuracy.
> >
> > ### RE: The experiments on the heterogeneous data distribution is not enough. The paper only uses synthetic data to simulate the heterogenous data distribution case. However, it would be more convincing if it could be done in the real-world data and it is also adopted by a lot of other federated learning methods to show their effectiveness. Partitioning the data with the dirichlet distribution would be a good way and the hyperparameter in the dirichlet distribution could be utilized to control the level of heterogeneity.
> >
> > Both real datasets we utilized are heterogeneous. We follow the same dataset implementations as [1] and present the details of the datasets in Table 2. The results on heterogeneous real datasets are presented in Figures 5-8.
> >
> > [1] Li, Tian, et al. "Federated optimization in heterogeneous networks." Proceedings of Machine learning and systems 2 (2020): 429-450.

---

### Comment · Action_Editor_MzF9 · 2024-06-13
**Final question before the decision**

Dear authors,

Looking at the eq. (6) and the central update step after it, it seems to me that the upcycled method is effectively approximating the gradient at $\overline{\omega}^{2m-1}$ using the difference $\overline{\omega}^{2m-1} - \overline{\omega}^{2m-2}$. So if the clients would share gradients instead of the parameters values, I wonder if you could establish the upscaled step, by simply using a larger step size at the odd steps?

---

> ### Author Response · Authors · 2024-06-16
> **To Action Editor MzF9**
>
> Thanks for your reply. As illustrated in Figure 1, our method can be viewed from two different perspectives: 1) applying first-order approximation to even updates effectively reduces information leakage by half; 2) the even updates can be reduced to a global aggregation view with a larger update step, akin to using gradients (parameter differences) from clients with a larger step size to update the global model.
>
> However, it is important to distinguish between odd and even steps. In FL, clients conduct several rounds of local updates and then send parameter values or parameter differences to the server for global updates. In even steps, we explicitly avoid accessing clients' data in some steps to reduce information leakage. If we simply use a larger step size and update the model with the same number of steps as others, they will incur the same information leakage (as client data is used in every step) and cannot improve the accuracy-privacy tradeoff. This is also verified in our experiments, where all baselines use the carefully-selected hyper-parameters with the best performance, and our upcycled-FL attains much better privacy-accuracy tradeoff than baselines (Fig. 3 and Fig. 4).

---

### Decision · Action_Editor_MzF9 · 2024-06-17

**Recommendation:** Accept with minor revision

**Comment:**

After reading the paper, I noticed some minor points of improvement I will list below:
- typo: "datatsets"
- I wonder if the $\approx$ in eq. (5) should be $=$? Or if it truly is an approximation, it should be made clear where it comes from.
- "selected clients got updated", I guess missing "which"?
- "we apply our Upcycled-FL method to six" I guess it's "seven" now
- "Femnist" >> "FEMNIST", caption of table 1
- Privacy guarantees need to be clearly communicated in Fig. 4. I would also suggest mentioning that the training is non-private in Table 1 and Fig. 2
- Report, what is the $\pm$ reported in Table 1. Similarly what are the errorbars depicting in Figs. 2, 3 and 4.

**Audience:**

Federated learning, and its privacy-preserving variants are important for the machine learning, and especially trustworthy machine learning community. Therefore, I believe the paper is of interest for the broad audience of TMLR. This sentiment was also shared by all the reviewers.

**Claims And Evidence:**

This paper proposes an alternative update schedule for federated learning algorithms. By using first order approximation of the loss function, the proposed method allows to reduce the data accesses between central update rounds by half. This allows to improve both the privacy guarantees and the communication efficiency of FL algorithms.

The method is tested with both synthetic and real data sets, with various different FL algorithms and learning tasks. The results demonstrate the effectiveness of the proposed method, thus providing empirical evidence for the claims.

---

> ### Author Response · Authors · 2024-07-19
>
> We greatly appreciate the time and effort all reviewers and the action editor have taken to review our paper. We have updated our paper and addressed the final comments:
>
> > typo: "datatsets"
>
> Thanks for reminding. We have corrected the typo "datatsets" to "datasets".
>
> > I wonder if the $\approx$ in eq. (5) should be $=$? Or if it truly is an approximation, it should be made clear where it comes from.
>
> Eq. (5) is the estimated update from the odd update, where the approximately equals sign "$\approx$" is due to the approximation in Eq. (2). We have added an explanation of its origin to make this clear.
>
> > "selected clients got updated", I guess missing "which"?; "we apply our Upcycled-FL method to six" I guess it's "seven" now; "Femnist" >> "FEMNIST", caption of table 1
>
> Thank you for the reminder. You are correct. We have fixed them in the camera-ready revision.
>
> > Privacy guarantees need to be clearly communicated in Fig. 4. I would also suggest mentioning that the training is non-private in Table 1 and Fig. 2
>
> We have updated Figure 4 to clearly communicate the privacy guarantees. Additionally, we have specified in Table 1 and Figure 2 that the training is non-private.
>
> > Report, what is the $\pm$ reported in Table 1. Similarly what are the errorbars depicting in Figs. 2, 3 and 4.
>
> We have specified that the $\pm$ indicates the standard deviation of the accuracy over four runs. We have also provided explanations for the error bars depicted in Figs. 2, 3, and 4.